# A spatially anchored transcriptomic atlas of the human kidney papilla identifies significant immune injury in patients with stone disease

Victor Hugo Canela [1,12], William S. Bowen [2,12], Ricardo Melo Ferreira [2,12], Farooq Syed [3], James E. Lingeman [4], Angela R. Sabo [1,2], Daria Barwinska [2], Seth Winfree [1,5], Blue B. Lake [6], Ying-Hua Cheng [2], Joseph P. Gaut [7], Michael Ferkowicz [2], Kaice A. LaFavers [2], Kun Zhang [6], Fredric L. Coe [8], Elaine Worcester [8], the Kidney Precision Medicine Project*, Sanjay Jain [9] ✉, Michael T. Eadon [2,10] ✉, James C. Williams Jr. [1] ✉ & Tarek M. El-Achkar [1,2,11] ✉

Kidney stone disease causes significant morbidity and increases health care utilization. In this work, we decipher the cellular and molecular niche of the human renal papilla in patients with calcium oxalate (CaOx) stone disease and healthy subjects. In addition to identifying cell types important in papillary physiology, we characterize collecting duct cell subtypes and an undifferentiated epithelial cell type that was more prevalent in stone patients. Despite the focal nature of mineral deposition in nephrolithiasis, we uncover a global injury signature characterized by immune activation, oxidative stress and extracellular matrix remodeling. We also identify the association of MMP7 and MMP9 expression with stone disease and mineral deposition, respectively. MMP7 and MMP9 are significantly increased in the urine of patients with CaOx stone disease, and their levels correlate with disease activity. Our results define the spatial molecular landscape and specific pathways contributing to stone-mediated injury in the human papilla and identify associated urinary biomarkers.

The prevalence of kidney stone disease, or nephrolithiasis, is increasing in the US and around the world. Nephrolithiasis is associated with significant morbidity, impaired quality of life, and significant health care utilization. This disease is complex, with a multifactorial etiology influenced by genetic and environmental factors[1,2]. Despite decades of innovation and efforts by researchers to describe its pathophysiology, the precise mechanisms contributing to kidney stone formation remain poorly understood[3]. A key factor leading to this knowledge deficit is the paucity of data on the cellular and molecular makeup of the kidney papilla and its alteration with stone disease. In addition, the spatial distribution of various cell types and their association with mineral deposition (such as Randall's plaque) during stone disease is largely unknown but needed to understand the pathogenesis of stone formation.

Experimental studies in rodent models of crystal formation and mineral deposition suggest that stone disease may be driven by inflammation, oxidative stress, and osteogenic-like changes in the kidney papilla[4–7]. However, human data in stone patients are often

A full list of affiliations appears at the end of the paper. *A list of authors and their affiliations appears at the end of the paper. ✉e-mail: sanjayjain@wustl.edu; meadon@iupui.edu; jwillia3@iu.edu; telachka@iu.edu

limited[3]. There is some evidence to suggest the role of the immune system in the pathogenesis of stone disease. For example, proteins associated with immune cell activation have been discovered in proteomic studies of kidney stones from patients[8–13]. However, it remains unclear if the identified proteins are directly involved in the formation of CaOx stones or if they are simply a byproduct of non-stone-related events[14]. The enrichment of specific immune proteins can depend on the type of stone, such as the preponderance of neutrophil proteins in brushite compared to calcium oxalate (CaOx) stones[15].

Renal papilla samples from stone formers are very challenging to obtain as they require a biopsy during a nephrolithotomy surgical procedure, and their size is only a few millimeters. Molecular data from such specimens is more limited. In CaOx stone formers, Taguchi et. al. showed enriched gene expression of pro-inflammatory M1 macrophages by bulk microarray analysis of human papillary samples, thereby supporting an important role of macrophage activity in kidney stone disease[16,17]. However, the complexity of immune cell types, their distribution, and their spatial neighborhoods are not known in humans. It is also unclear if the underlying pathology in stone disease is limited to areas of plaque deposition or if it is more diffuse across the papilla. Technological advances such as single-cell and spatial transcriptomics, and large-scale high-resolution imaging, allow for the spatial definition of cell types and states based on transcriptomics and protein markers. These technologies can advance our understanding of the pathogenesis of nephrolithiasis by defining spatial niches of various cell types/states in stone disease and addressing existing gaps.

In this work, we procured difficult-to-obtain human kidney papilla biopsy specimens from stone formers and reference nephrectomy tissues to create a spatially anchored transcriptomic atlas of the renal papilla by integrating single nuclear RNA sequencing (snRNAseq) and spatial transcriptomic (ST). By using high-resolution large-scale multiplexed 3D and Co-Detection by indexing (CODEX) imaging, we defined the spatial localization and niches of specific cell subtypes of the human papilla and the changes in this landscape with stone disease. We discovered that areas of mineral depositions are immune active zones consisting of immune injury and matrix remodeling genes that affect multiple cell types extending beyond areas surrounding mineralization. Our studies also identified MMP7 and MMP9 as potential urinary biomarkers associated with stone disease and its activity.

## Results

### Identification and localization of cell types in the kidney papilla

We recently reported a comprehensive single nuclear RNA sequencing (snRNAseq) profile (over 200,000 cells) of the adult human kidney from the Human Biomolecular Atlas (HubMAP) and the Kidney Precision Medicine project (KPMP) consortia[18]. We leveraged this dataset to gain a deeper understanding of cell type diversity in the human kidney papillae and unique features compared to the cortical-medullary cell types. Renal papilla samples from the original dataset were bolstered by additional samples from this work for a total of 76,351 nuclei representing the human papilla (Fig. 1a–f). A list of abbreviations is presented in Supplementary Table 1. These samples were acquired from papillary biopsies of CaOx stone formers or were dissected from reference nephrectomies. In addition to the expected papillary cell types such as principal (PapPC) and intercalated (IC) cells of the inner medullary collecting ducts, descending and ascending thin limbs, papillary surface epithelial and endothelial cells, we also identified a sizable number of stromal and immune cells (Fig. 1b, d). We also found a unique population of undifferentiated cells that were enriched in injury-associated genes and lacked a transcriptomic signature specific to a unique cell identity (Fig. 1d, e). The proportions of PapPC and fibroblasts in the papilla were the highest compared to the cortex and medulla (Fig. 1b).

Notable differences in gene expression were observed between papillary and cortico-medullary principal cells (Fig. 1c). For example,

the expression of urea transporter UT2 (SLC14A2) and Aquaporin 2 (AQP2) were high in papillary PCs. In contrast, the expression of the epithelial sodium channel ENaC (SCNN1G, SCNN1B) and PTH receptor (PTH2R) were more abundant in the cortico-medullary nuclei. The gene RALYL, encoding the RNA-Binding Raly-Like Protein and responsible for the cystic Bardet-Biedl Syndrome 1 (BBS), was predominantly expressed in PCs outside the papilla (Fig. 1c), aligning with the known cortical distribution of cysts in BBS[19]. Distinct sodium bicarbonate transporters were also expressed in papillary (SLC4A7) as compared to cortico-medullary cells (SLC4A4). These findings define known and novel gene expression profiles consistent with the physiological role of these papillary cells in regulating water and urea transport and acid-base balance in one of the highest regions of solute concentration in the human body. The snRNAseq analysis identified IC cells in the papilla, but they were less abundant than in the cortex (Fig. 1b). We confirmed their presence in the papilla by single molecular fluorescence in situ hybridization (smFISH) (Supplementary Fig. 1). Papillary IC cells have also notable differences in gene expression compared to the cortex (Supplementary Fig. 1a). IC cells in the papilla commonly express AQP2, a PC marker gene, suggesting that most papillary IC cells exhibit features of a "hybrid" IC-PC phenotype (Supplementary Fig. 1b–d). This "transitional" cell state has also been described in single cell RNAseq data from mouse and human kidneys[20–22]. In contrast, most IC cells in the cortex and medulla do not express AQP2 (Supplementary Fig. 1b, c), thereby supporting that anatomical location and the microenvironment influence cell specialization and gene signature profile.

Further characterization of the principal cells in the papilla showed that in addition to the healthy PapPC1 population, a PapPC2 population enriched with stress/injury genes was identified (Fig. 1f). These principal cell subtypes were compared to the undifferentiated epithelial cell type that also exhibited an injury signature. Both PapPC1 and PapPC2 cells expressed canonical papillary PC markers; however, PapPC2 cells also expressed injury markers such as VIM, JUNB, JUND, LCN2, and MAP1LC3B (LC3) (Fig. 1f). The undifferentiated epithelial cells had greater expression of injury markers such as PROM1, IGFBP7, and SPP1 and likely represent a degenerative state.

To validate these cell types and ascertain their spatial location in the papilla, we used spatial transcriptomics (ST)[23]. We mapped the snRNA-seq labels to resolve the ST gene expression profiles obtained from 10X Visium in reference and CaOx stone disease specimens (Supplementary Table 2 and Supplementary Fig. 2). The integrated analysis showed the mapping of the appropriate histological structures with the papillary cell types/ tubules with signatures identified by the snRNAseq (Fig. 1g–o, Supplementary Fig. 3). For example, PC cell signatures mapped to histologically identifiable CDs (Fig. 1i, j) and the majority of papillary surface epithelial cells (PapE) mapped to the outer edge as expected (Fig. 1k, l). Fibroblast and immune signatures were identified in all the samples tested (Supplementary Fig. 2). Importantly, the signatures of PapPC1, PapPC2, and undifferentiated cells were identified and mapped in the tissue. PapPC1 and PapPC2 were frequently co-localized to the same collecting ducts, suggesting that these various subtypes of PC are not spatially segregated in different collecting ducts, and may represent different cell states. We used smFISH to validate the presence of these unique cell types (Fig. 1p, q and Supplementary Figs. 4 and 5) using the markers from a differential expression analysis of the snRNAseq (Fig. 1f). PapPC2 cells in the collecting ducts have high expression of MMP7 and IGFBP7 compared to PapPC1 cells (Fig. 1p and Supplementary Fig. 4). Undifferentiated cells often localize to the interstitial space or to tubules morphologically consistent with thin limbs. They have high expression of IGFBP7 and frequently co-express PROM1 suggesting an altered degenerative state[18] (Fig. 1q and Supplementary Fig. 5).

To extend the transcriptomic cell type prediction to the protein level, we performed highly multiplexed CODEX imaging of reference

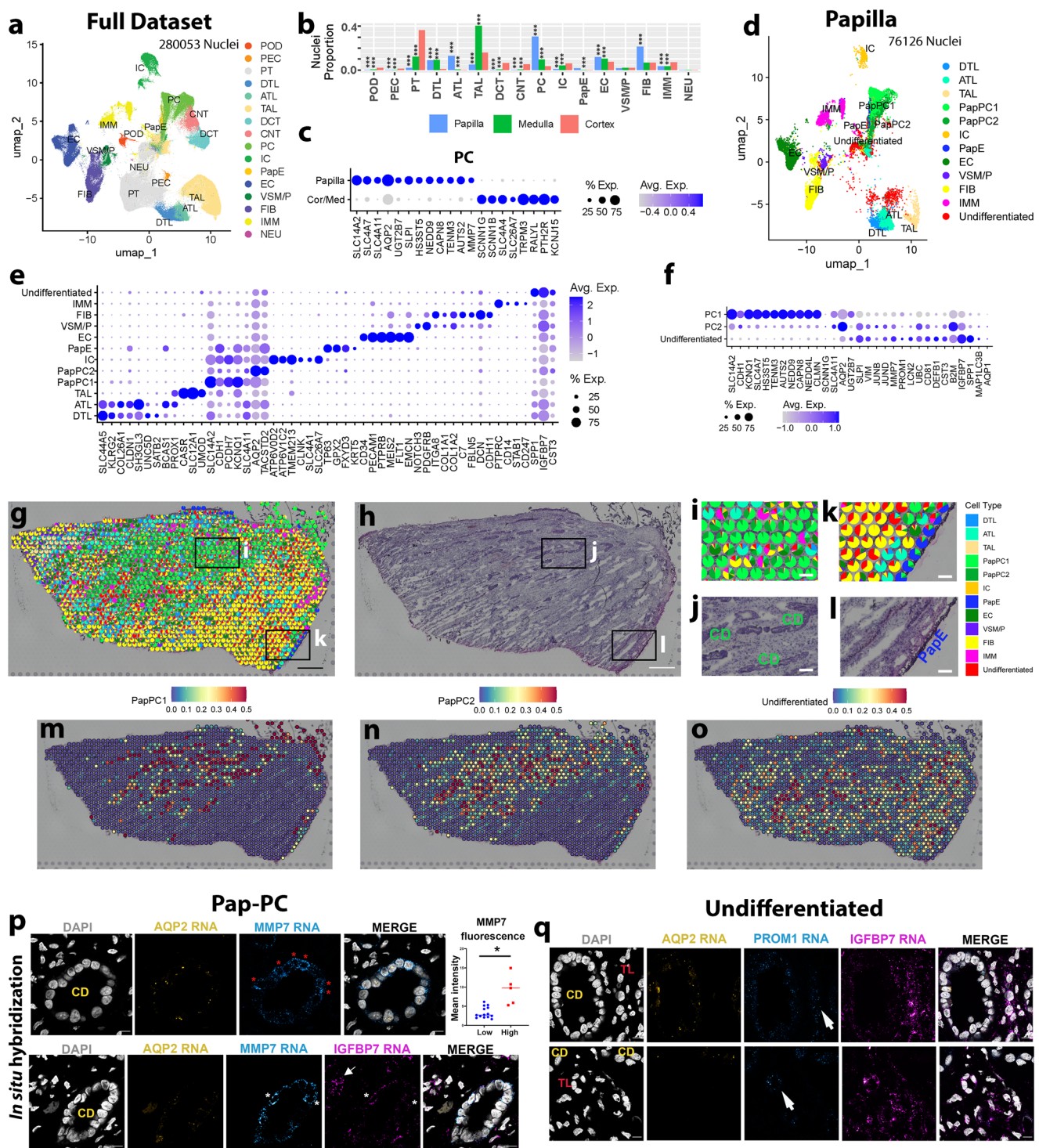

tissues using 32 cell markers (Supplementary Table 3 and Fig. 2a, b). CODEX methodology and quality control measures are detailed in the methods section and Supplementary Figs. 6 and 7). Applying unsupervised analysis and classification based on the markers used, the presence of various expected cell types was confirmed and mapped within the tissue (Fig. 2c, d, Supplementary Figs. 7 and 8). Most of the cell types aligned with the cell types from the transcriptomic datasets with some overlapping markers from Fig. 1f. We confirmed the presence of a significant resident immune cell population, with a predominance of macrophages expressing CD206 (CD68 low, CD206 + ) (Fig. 2c, d) consistent with recent data from McEvoy et al.[24]. CODEX analyses uncovered two populations of collecting duct cells in the

papilla (Fig. 2c–f), CD-1 and CD-2, which predominantly consist of principal cells based on the distribution of papillary collecting cell types outlined in Fig. 1b. These two populations were differentiated by the expression of injury and renewal markers in CD-2 such as LC3, p-MLKL and Ki67 (Fig. 2f, Supplementary Fig. 8) and consistent with the transcriptomic injury signature uncovered in PapPC2 cells (Fig. 1f). The existence of 2 populations of collecting duct cells at the protein level was further validated using confocal immunofluorescence (IF) for MMP7 and AQP2 (Supplementary Fig. 9), which directly overlap with the transcriptomics and smFISH analyses. Using CODEX, we also identified a population of undifferentiated epithelial cells (Fig. 2e, f) whose protein expression profile overlaps with the undifferentiated

**Fig. 1 | Spatially anchored cellular and molecular characterization of the human kidney papilla. a** Integrated KPMP/HuBMAP snRNAseq atlas combining nuclei from the renal cortex, medulla, papilla, with additional local papilla samples. **b** The proportion of cell type representation among nuclei by kidney region. *** denotes P < 0.0001 using two-tailed Fisher's exact test compared to cortex. **c** Differential gene expression analysis of papillary principal cells (PC) reveals increased urate 2 transporter (SLC14A2) expression as compared to cortical or medullary PCs. **d** Subset atlas of nuclei specific to the papilla. **e** Gene expression profile of the papillary cells from (**d**). **f** Gene expression signatures of PapPC1, PapPC2, and undifferentiated cells in snRNAseq data, showing a spectrum of injury in the PapPC2 and undifferentiated cell types. **g** Label transfer of snRNAseq cell classes onto spatial transcriptomic spots within a reference papilla tissue, with underlying histology shown in (**h**). **i, j** An enlarged area denoted by the box in (**g** and **h**) showing PCs mapping on histologically identified collecting ducts. **k, l** An enlarged area denoted by the boxes in (**g** and **h**) showing papillary epithelial (Pap-E) cells mapping on histologically identified papillary epithelium. **m–o** Feature plots showing the mapping of Pap-PC1, Pap-PC2, and undifferentiated cell signatures. Pap-PC1 and Pap-PC2 frequently overlay collecting duct (CD) histology. **p** Single molecule fluorescence in situ hybridization (smFISH) for cell markers from (**f**) to distinguish PapPC1 and PapPC2. Collecting ducts (CD) are morphologically distinct and express Aquaporin 2 (AQP2). Red asterisks denote cells with high MMP7 expression (Pap-PC2). MMP7 mean fluorescence intensity in these cells ($n = 5$) was significantly higher than other cells ($n = 14$) in that CD. (*$p = 0.0001$) by unpaired two-tailed t-test. The lower panel in (**p**) shows that few Pap-PC2 cells (white asterisks) also express IGFBP7, consistent with data in (**f**). The arrow shows undifferentiated cells outside of CD with high IGFBP7. **q** smFISH highlights undifferentiated cells frequently co-expressing IGFPB7 and PROM1 (arrows). These cells localize to the interstitium or in cells with morphology of thin limbs (TL). Extended data is shown in Supplementary Figs. 4 and 5. Scale bars: 0.5 mm in (**g** and **h**), 100 μm in (**i–l**), and 10 μm in (**p–q**).

population uncovered by transcriptomics (high expression of IGFBP7 and PROM1). Thus, using snRNAseq, ST, smFISH, CODEX, and confocal IF, we validated the cellular diversity in the papilla and identified new cell populations with plausible biological significance (see below).

Using CODEX, we explored the population of stromal cells localized in the papillary interstitium, which was divided into separate cell classes based on an unsupervised analysis (Fig. 2g–i). We uncovered and spatially mapped two populations of fibroblasts based on the expression of fibronectin (FN1), vimentin (VIM), and osteopontin (SPP1). We also identified a population of myofibroblasts with high expression of α-SMA (Fig. 2i). The presence of fibroblasts in the papilla is consistent with our snRNAseq and ST findings in Fig. 1 and Supplementary Figs. 2 and 3.

### Transcriptomic signatures of CaOx stone formation within the renal papilla

Next, we used the snRNAseq data to compare stone and refence tissue (Fig. 3). As shown in the UMAPs in Fig. 3a, the undifferentiated cell cluster was remarkably expanded in stones samples. When comparing at the patient level (Fig. 3b), the proportion of undifferentiated cells was significantly increased in stone vs. reference samples ($p = 0.03$). We then determined the differentially expressed genes (DEGs) between CaOx stone and reference samples for each cell type in the papilla (Fig. 3c, Supplementary Fig. 10). Multiple cell types in stone samples expressed increased levels of injury and stress genes such as SPP1, CLU, LCN2, S100A11 and MMP7 (Fig. 3c, d, Supplementary Fig. 10). Pathway analysis detected significant enrichment of common pathways related to protein translation, among the cells from patients with stone disease. We also observed the enrichment of pathways of interest (Fig. 3c and Supplementary Fig. 11), such as immune system process/ leukocyte activation, response to oxidative stress, ossification, and extracellular matrix organization in most cell types.

ST of stone samples showed that undifferentiated cells frequently localized near areas of mineralization and are likely a result of localized injury to the adjoining tubules (Fig. 3e–h). When examining changes in the distribution of the gene expression signatures, we observed an increase of immune, fibroblast and undifferentiated cell signatures and a concomitant decrease in tubular cell signatures (Fig. 3i) The injury genes observed in snRNAseq were similarly upregulated in the analysis of global gene expression of ST stone samples vs. reference (Fig. 3j). Enriched pathways obtained independently from the spatial transcriptomic analysis overlapped with the snRNAseq pathway analysis (Fig. 3c, j, Supplementary Fig. 11). The concordant results between technologies increase the confidence in the biological relevance of these pathways in the pathogenesis of the stone disease.

To further gain insights into the pathogenesis of injury in stone disease, we looked in more detail into MMP7 expression, a gene involved in injury and matrix remodeling and consistently upregulated in the papillae from patients with stone disease. Supplementary Fig. 12

shows the spatial and relative gene expression maps of *MMP7*. Interestingly, in reference tissue, *MMP7* was confined to CDs, but in stone patients *MMP7* showed a diffuse expression pattern, which was consistent with the generalized increased expression observed in most cell types from the snRNAseq and combined analysis (Fig. 3d, j and Supplementary Fig. 10). The higher level of MMP7 expression in collecting ducts was validated using immunofluorescence confocal microscopy staining for MMP7 and AQP2 (Supplementary Fig. 12e). Analysis of MMP7 expression using confocal 3D imaging and tissue cytometry (as detailed in Supplementary Fig. 9) shows that the proportion of collecting duct cells with high MMP7 expression (likely corresponding to PapPC2) is increased with stone disease Fig. 3k, which is consistent with the transcriptomic data. Cumulatively, our data suggest stone disease alters the phenotype of collecting duct and other tubular cells toward an injury phenotype. These changes may trigger immune and fibrogenic responses that eventually replace the healthy tubular epithelial make-up of the papilla.

### Regional analysis of mineralized and non-mineralized tissues

Mineral deposition such as Randall's plaque (RP) formation is thought to play an important role in the formation of stone disease. However, plaque formation is frequently focal, and the cellular and molecular alterations present in the microenvironment of mineral deposits could be important in the pathogenesis of this disease. Supervised analysis of spatial transcriptomic profiles based on the selection of mineral deposits in a stone forming papilla are shown in Fig. 4. The results indicate that transcript signatures of genes such as *NEAT1, CHIT1, LYZ,* SPP1, and *MMP9* are upregulated in areas contiguous to mineralized tissue compared to regions more distant (non-contiguous) to mineral (Fig. 4). These genes are also associated with pathways of leukocyte activation and response to oxidative stress (Supplementary Fig. 13). The expression of genes involved in macrophage activation such as *MMP9* and *CHIT1* (Fig. 4b) was upregulated in the area of mineralization as compared to the non-contiguous region. The high expression of SPP1 in areas of mineralization was also validated at the protein level using CODEX imaging (Fig. 4j–o).

We next sought to better define the niche of the mineralization nidus and understand its contribution to injury, particularly with our findings of enrichment of immune cell marker genes and their association with mineral deposition. We used CODEX multiplexed imaging in conjunction with a panel of antibodies directed to different immune cell types (Fig. 5). We took advantage of the autofluorescence properties of mineral deposits (Fig. 5a), whereby the plaque can be identified without staining[15,25]. The main cell clusters identified in the reference tissue (Fig. 2c, d) were also identified here, except that there was marked expansion of the immune clusters (Fig. 5b). Additional unsupervised analysis (Fig. 5c) was performed on the CD45+ cell clusters from the initial analysis (Fig. 5b), which resolved immune cells into major lymphocyte and myeloid clusters. The latter can be broadly

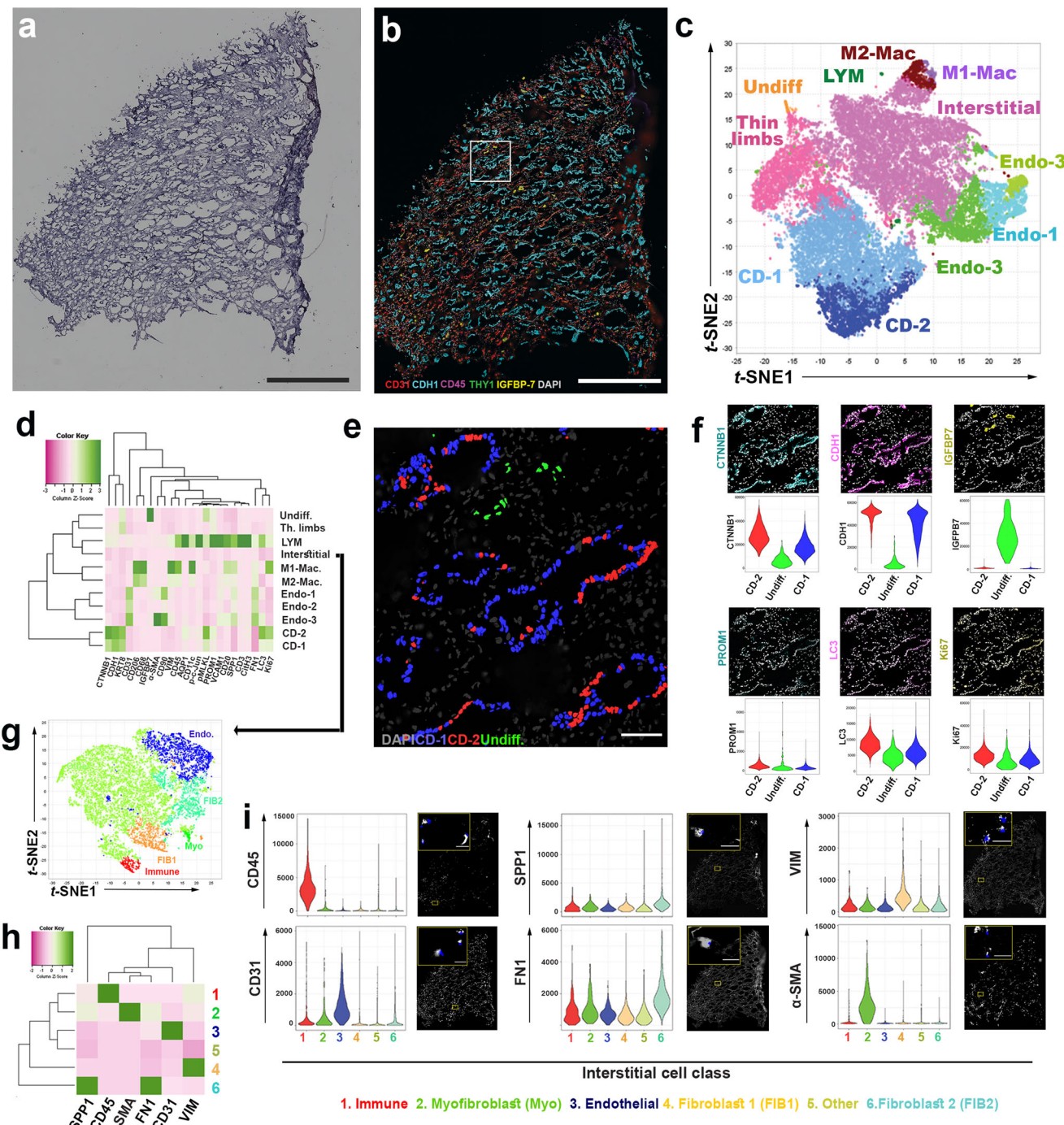

**Fig. 2 | Co-detection by indexing (CODEX) multiplex imaging of a reference papilla tissue sample. a** H&E stained section that underwent CODEX imaging showing characteristic papillary morphology. **b** CODEX multiplex imaging displaying 6 markers (CD31- endothelial cells; Cadherin-1 (CDH1)- in papilla marks collecting ducts; CD45- pan immune marker; THY1 (also known as CD90)- marks pericytes; IGFBP7- Injury marker and DAPI- labels nuclei). **c** Unsupervised clustering and dimensionality reduction of the CODEX data in a t-stochastic neighborhood embedding (t-SNE) plot showing various cells classes, which were validated by the level of fluorescence intensity (heatmap in (**d**) and examining levels of fluorescence in (**f**)) and mapping back on the image (example in (**e**) for CD cells and more details also in Supplementary Figs. 7 and 8). **e** localization of CD-1, CD-2 and

undifferentiated cells (Undiff.) in CODEX images using colored nuclear overlays. **f** shows the corresponding staining of various markers that characterized the profile of CD-1, CD-2 and Undiff. cells with the corresponding distribution of fluorescence intensity for these markers. CD-2 cells express higher levels of injury markers but are not segregated into separate tubules. Undifferentiated cells are localized to interstitium. **g** Re-clustering of interstitial cells based on specific markers **h** in CODEX data identifies populations of fibroblasts (2 subtypes, FIB1 and FIB2) and myofibroblasts (Myo), which were mapped into the interstitium as shown in panel **i** for each marker. Scale bars: (0.5 mm in **a**, **b**, 0.1 mm in **e/f** and 0.02 mm in (**i**)). Other abbreviations listed: Endo = endothelial cells; LYM = lymphocytes; Th. Limbs = thin limbs.

divided into three categories based on markers such as CD68, CD11c, and CD206, which correspond to an inflammatory (M1, CD68 high[26–28] and CD11c high[29,30]), alternatively activated (M2, CD68 low, CD206 +) and intermediate phenotype (M1/M2, CD68 + , CD206 + ). Mapping

back immune cells to the tissue images reveals that certain areas of plaque deposition serve as a nidus for immune activation (Fig. 5d), where the deposited mineral is surrounded by inflammatory macrophages (M1 and M1/M2), which are known to be antigen-presenting

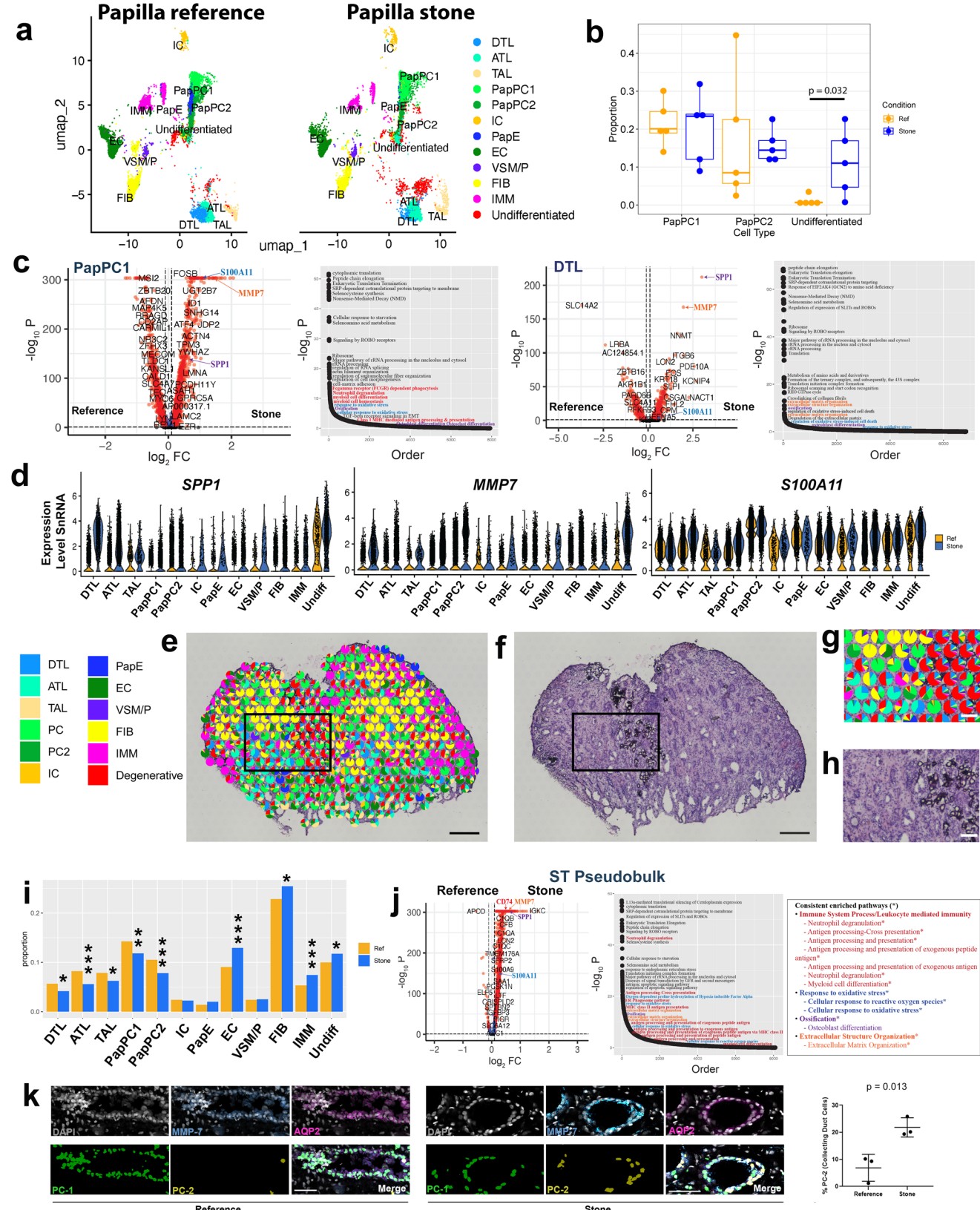

cells (APCs) (Fig. 5e). At the periphery of this immune nidus, these APCs appose CD4 + T and B cells, a phenomenon consistent with immune activation (Fig. 5e). Interestingly, not all the plaque areas are involved with immune activation, and neighboring areas with mineral did not have significant immune cells. We then examined the distribution of other interstitial cell populations such as fibroblasts and

myofibroblast in these areas (Fig. 5f, g). The plaque areas without heightened immune activity had significant fibroblast infiltration (Fig. 5h). Cumulatively, these data demonstrate that the presence of plaque is immunogenic. Mineral deposition likely elicits a varied immune response with a defined progression from myeloid to a mixed lymphocytic infiltrate and a non-overlapping fibrogenic response

**Fig. 3 | Differentially expressed genes (DEGs) and cell states induced by stone disease within the human papilla. a** UMAPs display cell types clustered from snRNA sequencing in papillary reference and stone samples, highlighting the expansion of the undifferentiated cell population in stone disease. **b** Proportion of PapPC1, PapPC2 and undifferentiated cell types for each specimen (n = 5 stone and n = 5 reference independent papillary specimens), validated the expansion of the undifferentiated population (box plots display median and interquartile range; two-tailed Wilcoxon rank sum test was used). **c** DEGs (two-tailed Wilcoxon rank sum test) and enriched pathways (overrepresentation test, see methods) in papillary cells by snRNAseq, focusing on PapPC1 and descending thin limbs (DTL, a cell important in stone disease pathogenesis). DEGs for other cell types are shown in Supplementary Figs. 10–11. Pathway analysis detected significant upregulation of ossification (purple), extracellular matrix organization (orange), response to oxidative stress (blue) and immune system process /leukocyte activation pathways (red text). **d** Expression of SPP1, MMP7 and S100A11 are consistently induced in stone disease across papillary cell types. **e** Label transfer of cell classes onto spatial transcriptomic spots in a stone disease papilla with underlying H&E histology (**f**). **g** An enlarged area from (**e** and **f**) showing the undifferentiated cell signature localizing to areas of mineralization (**h**). The distribution of cell signatures (percent of total spots for reference vs stone disease) in ST specimens is shown in (**i**). *p < 0.01, **p < 0.001 and ***p < 0.0001 (two-tailed Fisher's exact test). **j** DEGs between pseudobulk (combined gene expression of spots) of reference (left) and CaOx stone ST samples (two-tailed Wilcoxon rank sum test). Differentially enriched pathways (overrepresentation test) in stone disease are illustrated in the pathway curve. Consistently enriched pathways that were common in each individual sample (N = 6) compared to reference samples (N = 4) are listed. **k** Analysis of MMP7 protein expression using 3D imaging and tissue cytometry (N = 3 reference and N = 3 stone specimens), showing a higher proportion of MMP7 high cells (PapPC2) within collecting ducts in stone disease vs. reference (mean ± standard deviation (error bars); p = 0.041 -unpaired two-tailed t-test). Scale: **e**, **f** = 300 μm; **g**, **h** = 100 μm; **k** = 10 μm.

---

consisting of activated fibroblasts and myofibroblasts for repairing the injured tissue.

## Large scale 3D imaging and tissue cytometry establishes inflammatory stress signaling and macrophage activation as cardinal features of stone disease

Next, we wanted to determine if immune activation and oxidative injury are cardinal features in the papilla of CaOx stone-forming patients, independent of large mineral deposits. Using quantitative large-scale 3D imaging and tissue cytometry on papillary tissue specimens from controls and subjects with CaOx stone disease and no visible mineral deposits (N = 4 each group), we quantified the expression and distribution of phosphorylated c-JUN (p-c-JUN, marker of oxidative stress and stress kinase activation) and CD68 (marker for macrophages and inflammation) (Fig. 6). Patients with CaOx stones compared to controls had a significantly higher abundance of CD68+ macrophages (3.4 +/− 0.8 vs. 1.2 +/− 0.7 % of total cells, respectively; p = 0.0103) and p-c-JUN+ cells (12.4 +/− 5.2 vs. 4.0 +/− 2.4 % of total cells, respectively, p = 0.0279) than non-stone formers. Macrophage infiltration was diffuse, and activation of c-JUN was not restricted to a specific cell type (Fig. 6d), which is consistent with the findings from snRNAseq and transcriptomic analyses suggesting increased stress signaling across multiple cell types in the papillae of stone patients.

## MMP7 and MMP9 levels are increased in urine of kidney stone patients and correlate with disease activity

The snRNAseq and ST datasets revealed upregulation of MMP7 gene expression in the papillae of patients with stone disease and the upregulation of MMP9 gene expression in regions of mineralization. While these two matrix remodeling genes could explain some of the mechanisms behind papillary stone-associated injury, we next asked if their urinary secretion can also screen for stone disease activity. To this end, we assayed for MMP7 and MMP9 in urine sample from a cohort of 55 patients with normal kidney function separated into 3 groups (Fig. 7): 1) healthy controls with no known clinical history of stones, 2) inactive CaOx stone formers with a known clinical history of stones but without a recent stone event and 3) active CaOx stone formers undergoing surgery for stone removal. Demographic and relevant clinical characteristics of these subjects are presented in Supplementary Table 4. Our results show that the levels of MMP7 are increased in the urine of CaOx stone patients without active disease compared to healthy subjects (4.8 ± 6.1 vs. 1.6 ng/mg Cr, respectively; p = 0.01). MMP9 was also significantly higher in the urine of inactive stone patients compared to healthy controls (1.8 ± 3.7 vs. 0.18 ± 0.17 ng/mg Cr, respectively; p = 0.02). In active stone formers, we observed even higher levels of urinary MMP7 (8.1 ± 6.6 ng/mg Cr; p = 0.0004 vs. controls and p = 0.47 vs. inactive stone formers) and MMP9 (8.0 ± 10.7 ng/mg Cr; p < 0.0001 vs. controls and p = 0.01 vs. inactive

stone formers). These results suggest that both MMP7 and MMP9 are potentially important markers to monitor stone disease activity.

## Discussion

In this work, we utilized snRNA sequencing, spatial transcriptomics, single molecular fluorescence in situ hybridization and large-scale multiplexed imaging to establish a spatially anchored cellular and molecular atlas of the renal papilla in non-stone and kidney stone disease tissues. An important strength of our study is the utilization of rare and highly valuable papillary biopsy specimens obtained from stone patients. A full landscape of papillary cells was defined, including the presence of papillary surface epithelial cells, stromal and immune cells, unique subtypes of principal cells, and an undifferentiated epithelial cell type that localized to regions of injury or mineral deposition. Despite the focal nature of mineral deposition in stone disease, we showed that injury pathways are globally upregulated across multiple cell types within the papilla. Commonly enriched signaling pathways in stone disease such as immune system process/ leukocyte (myeloid) immune activation, response to oxidative stress and extracellular matrix organization were demonstrated using orthogonal approaches, thereby enhancing confidence in the findings. We defined the microenvironment of plaque as an active immune zone with antigen-presenting inflammatory macrophages interacting with T cells, but also demonstrated the presence of an immune lifespan around mineral deposition ranging from inflammation to fibrosis. Finally, MMP7 and MMP9 were identified as two proteins linked to active stone disease and mineralization within the papilla. The levels of MMP9 and MMP7 in the urine were significantly higher in patients with history of stone disease compared to healthy controls and correlated with disease activity.

The papilla was enriched for PapPC1 and PapPC2 cells, which frequently co-localized in the same collecting ducts. PapPC2 exhibited a transcriptomic signature of cell stress. IC cells and transitional IC-PC cells were also observed, consistent with previous reports[20–22]. Interestingly, SLC8A1 (also referred to as Na + /Ca+ exchanger 1 (NCX1)), which was found to be specific to transitional cells by Saxena et al.[22], was predominantly expressed by cortico-medullary ICs compared to Pap-ICs (Supplementary Fig. 1a), which suggests a unique molecular profile for collecting duct cells based on their location and physiological function. Another cell type of interest was the "undifferentiated" snRNAseq epithelial cell cluster. Its signature did not align completely with a specific epithelial cell type but exhibited gene expression features of injury and degeneration. We posit that this cluster may represent a final common injury phenotype derived from epithelial cells with multiple origins. This cell type was highly prevalent in the stone samples and was localized to areas of mineralization in spatial transcriptomics (ST). Spatial mapping with CODEX suggested that these cells are likely a mixture of injured thin limbs and papillary

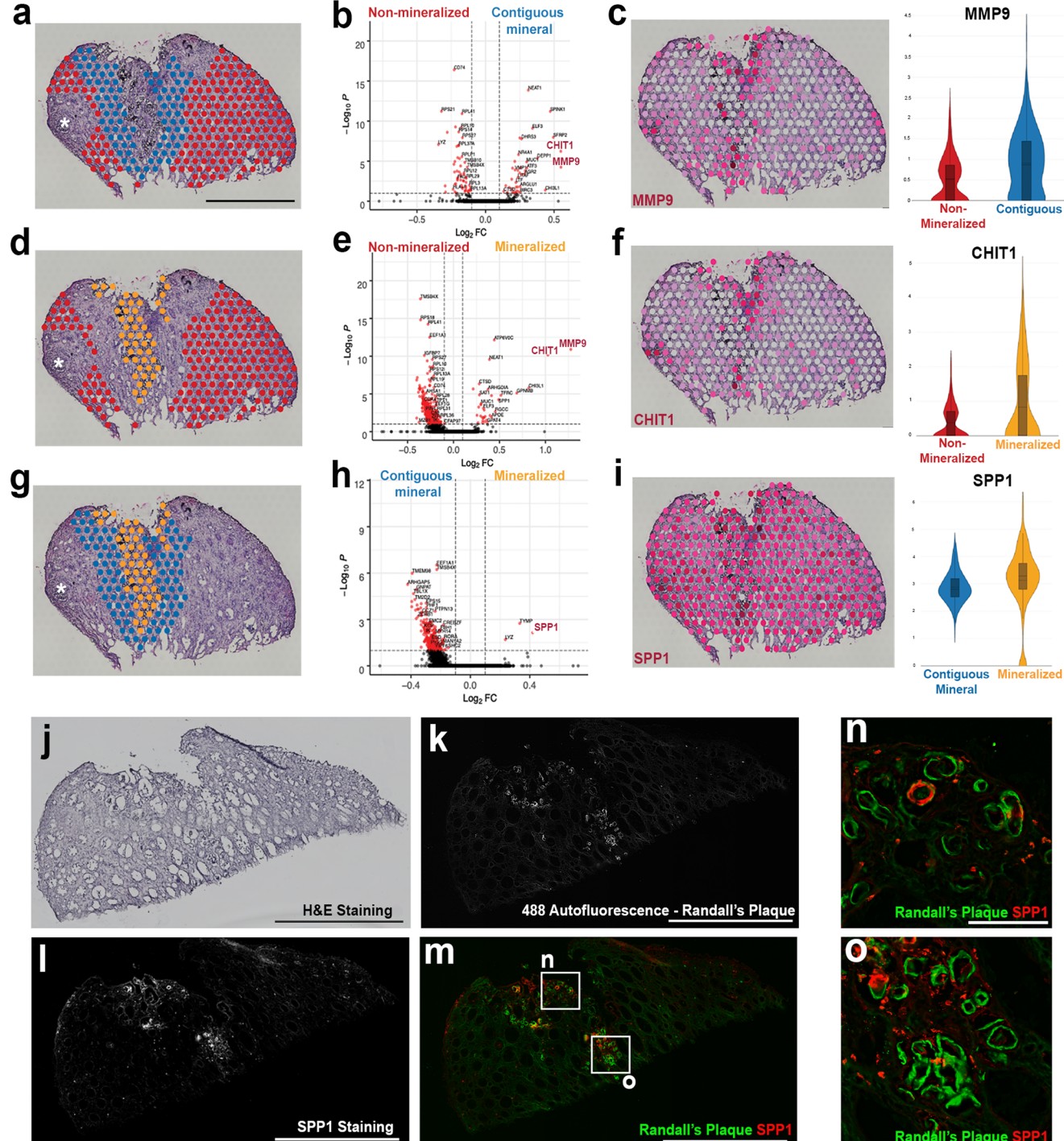

**Fig. 4 | Signatures of injury and inflammation are localized to regions of mineralization in a kidney papilla with stone disease. a** Regions of non-mineralization (*n* = 413 spots) compared to areas contiguous to mineral (*n* = 204 spots) display differentially expressed genes (DEGs) (by two-tailed Wilcoxon rank sum test) associated with pathways of leukocyte activation, such as *MMP9* and *CHIT1* (**b**). Scale = 1000 μm. *MMP9* expression localizes in areas contiguous to minerals and in regions of mineralization (**c**). **d** Comparisons between areas of non-mineralization (*n* = 413 spots) and areas of mineralization (*n* = 95 spots) also display DEGs (two-tailed Wilcoxon rank sum test) such as *MMP9* and *CHIT1* (**e**). *CHIT1* expression also appears to be more robust in areas of mineralization (**f**). Regional comparison between areas contiguous to mineral (*n* = 204 spots) and areas of mineralization (*n* = 95 spots) is shown in (**g**). SPP*1*, *TYMP*, and *LYZ* were differentially expressed (two-tailed Wilcoxon rank sum test) in

areas of mineralization in this CaOx biopsy specimen (**h**). SPP*1* appears to be localized to regions of mineralization but also displays relatively high expression throughout this stone-forming papilla (**i**). Violin plots show Log-Normalized values. Asterisks in **a**, **d**, and **g** denote that the analysis excluded the areas of mineralized plug and was restricted to Randall's plaque (RP). Box plots in **c**, **f** and **i** display the mean, median, interquartile and 95% range. H&E staining (**j**) corresponding to a papillary biopsy section with RP mineral deposition from a stone patient that was imaged using CODEX multiplexed imaging (**k**–**o** and Fig. 5). Scale = 1000 μm. 488 autofluorescence shows regions of RP (**k**). SPP1 staining is shown in (**l**). Overlayed images from (**k**) and (**l**) show association between RP and SPP1 (**m**). Scale in **k**–**m** = 1000 μm. Boxed regions highlighted in m are enlarged in (**n**) and (**o**) to show regions where RP and SPP1 co-localize. Scale = 10 μm.

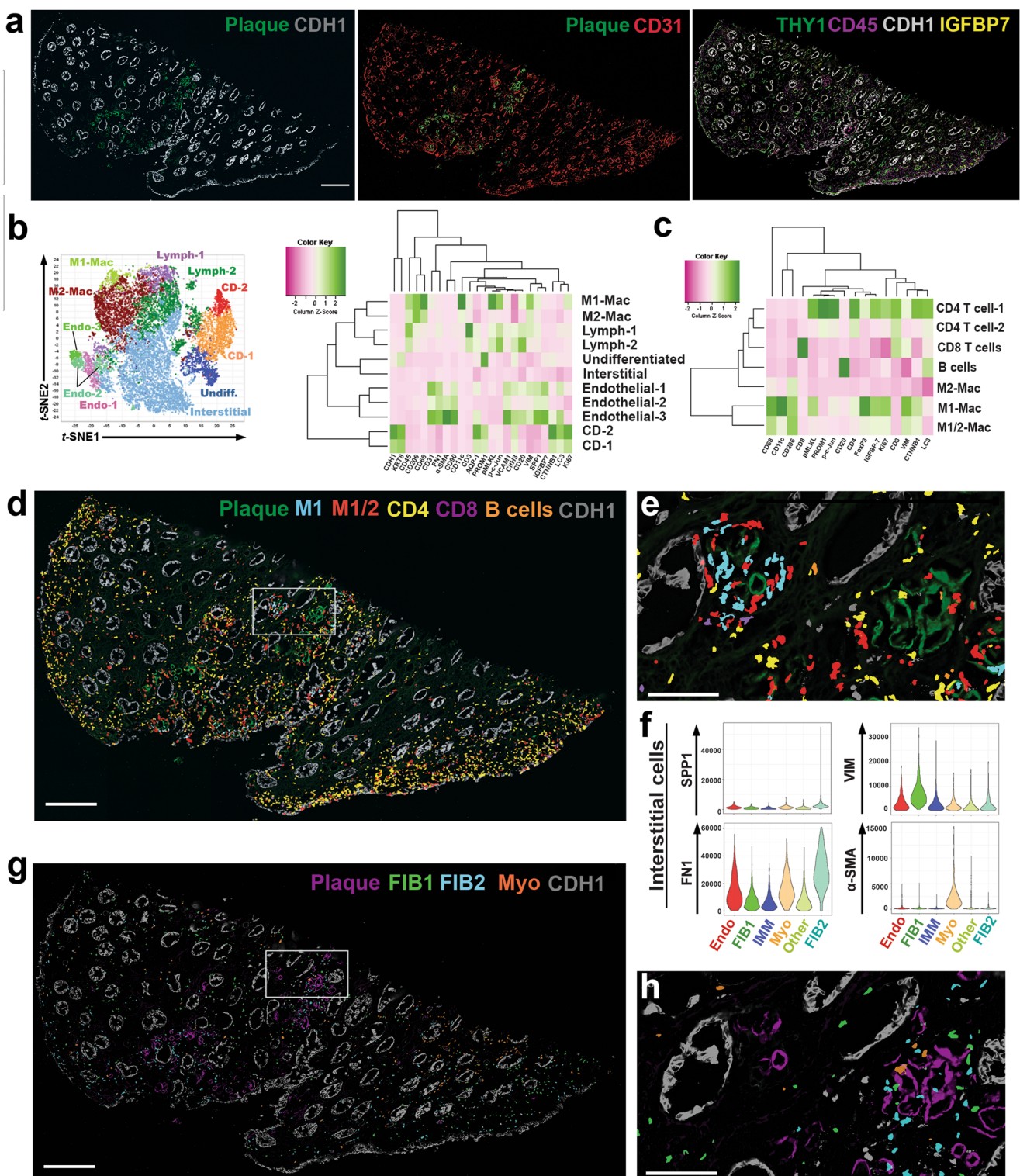

**Fig. 5 | CODEX imaging of a papilla with mineral deposition identifies various stages of immune activation and fibrosis around the plaque. a** CODEX imaging showing unique autofluorescence of the plaque, which can be easily delineated from the epithelial and vascular cells. **b** Unsupervised analysis and clustering identify similar clusters as in the reference specimen (Fig. 1), with extensive expansion of the immune clusters. **c** Re-clustering and analysis of the immune cells identifies all the major subtypes of leukocytes. Macrophages had an intermediate phenotype between M1 and M2 based on the co-expression of specific markers. **d** Mapping of immune cells in the tissue (using nuclear overlays) reveals niches of immune activity in certain plaque areas (area of mineral on left in (**e**), which is a high magnification view of the boxed area in (**d**)) with features of antigen presentation (interaction of antigen presenting macrophages with T and B cells) and a diffuse activated T cells response, particularly towards the papillary endothelium. Interstitial cells from (**b**) were re-clustered and analyzed (similar analysis as detailed in Fig. 2g, h) based on specific markers. **f** The levels of a few markers of interest are displayed based on interstitial cell class. Fibroblasts and myofibroblasts were abundant throughout the tissue (**g**), but fibroblasts were concentrated in certain areas of mineral deposition with reduced immune activity (area of mineral on right in (**h**), which is a high magnification view of the same boxed area in (**d**) and (**g**)), suggesting progression from inflammation to fibrosis in neighboring areas within the same tissue. Scales bars: 500 μm in (**a**), (**d**) and (**g**); 100 μm for (**e**) and (**h**).

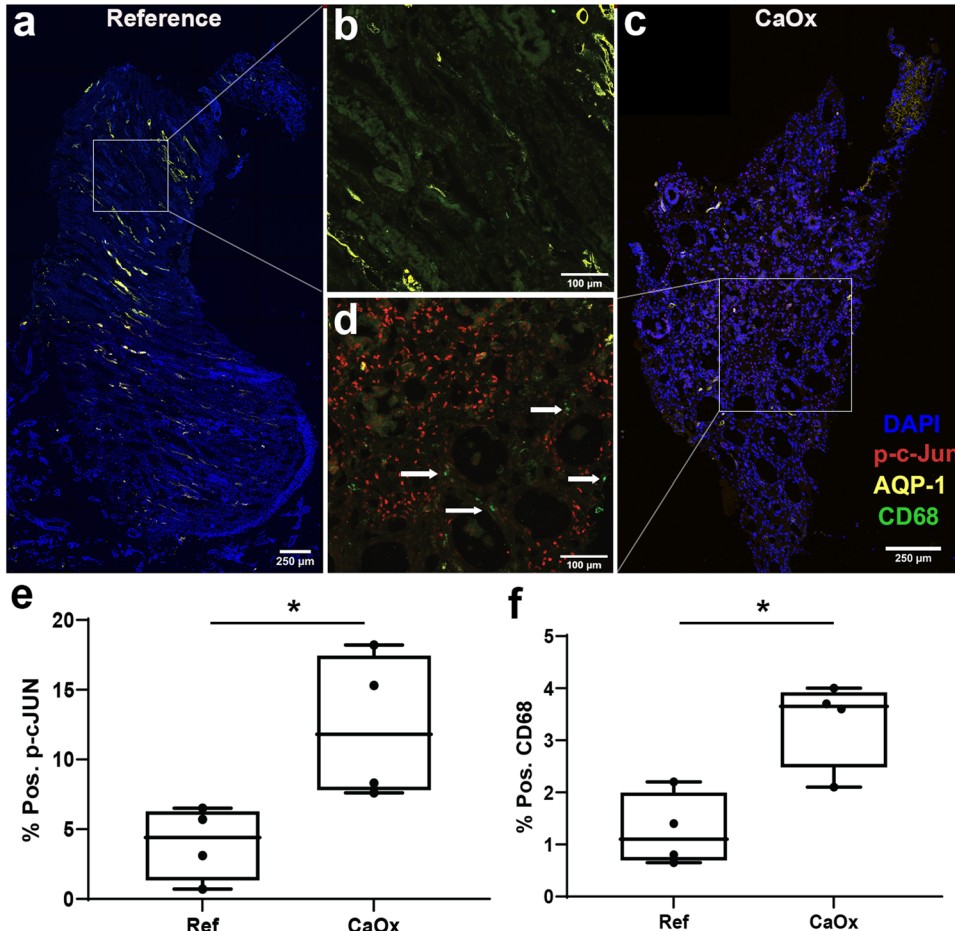

**Fig. 6 | Protein markers of oxidative stress (ROS) and macrophage activation are diffusely increased in biopsies of stone patients. a–d** Representative multi-fluorescence confocal images of kidney papillary biopsies from stone patients and reference nephrectomy tissue specimens (*n* = 4 stone and *n* = 4 refence independent tissue specimens) stained for phospho-c-JUN (p-c-Jun, marker of ROS), CD68 (activated macrophages) and Aquaporin1 (AQP-1, marker for thin descending limbs and descending vasa recta). Images were analyzed using volumetric tissue exploration and analysis (VTEA) software and the resulting outcomes are shown in (**e**) and (**f**) for p-c-JUN and CD68 as percentages of total cells in each tissue. Box plots show mean, interquartile range, minimum and maximum values. Boxed areas in (**a**) and (**c**) are enlarged in (**b**) and (**d**), respectively. Arrows in d shows CD68+ macrophages. Scale bars are 250 μm (**a** and **c**) and 100 μm (**b**–**d**). * *p* = 0.03 and 0.01 for reference vs. control for p-c-JUN and CD68, respectively, using a two tailed unpaired t-test.

epithelium, which is consistent with the location of these cells in the UMAP space (Fig. 3), and with smFISH spatial studies (Supplementary Fig. 5). A signature of injury in thin limbs is concordant with previous data from our group showing that Randall's plaque begins in thin limb cells[31]. This injured population and its potential association with pathology in the papilla needs further exploration.

Our data provide important human tissue context to the previously reported hypotheses, and some of the experimental models of stone formation which involved inflammation, leukocyte activation (macrophage activity), ROS, and ossification-like events[5,7,16,32–34]. Our spatial transcriptomic mapping of gene expression in stone disease agrees with, and extends the work of Taguchi and colleagues, who explored genome-wide analysis of gene expression on renal papillary Randall's Plaques (RP), and non-RP, and showed upregulation of *LCSN2, IL11*, and *PTGS1* in the RP patient tissue[7]. RP has been previously described as the interstitial mineral deposition at the tip of the renal papillae that can serve as the origin for CaOx stone growth[35,36]. This immune active state in the regions of papillary mineralization has a molecular profile comparable to vascular inflammation leading to atherosclerotic disease, which has been proposed to play a pathogenic role in mineral deposition and stone disease[37–40]. Our data suggest that papillary injury triggers pro-fibrotic signaling and a vigorous matrix remodeling program. Indeed, our studies uncovered two matrix metalloproteinase molecules, MMP7 and MMP9, that are associated with stone disease and mineral deposition, respectively.

Matrix metalloproteinases (MMPs) degrade extracellular matrix proteins during growth, tissue remodeling, and disease processes, and are secreted by various cell types including fibroblasts and leukocytes (including macrophages)[41]. Some MMPs control leukocyte migration and can modulate the immune response by biochemical cleavage of cytokines and chemokines[42]. MMP7, or matrilysin, is typically not expressed at the protein level in the renal cortex in healthy states[43,44] but its expression and activity are increased in the setting of kidney disease. There are no previous studies that have investigated MMP7 in the human papilla. In reference tissue, MMP7 expression was predominantly localized to collecting duct cells. The induced expression of MMP7 with nephrolithiasis could therefore be a valuable indicator of papillary injury associated with this disease.

In our studies, MMP9 (macrophage gelatinase) was associated with mineral deposition, which co-localizes with osteopontin (SPP*1*) expression. *MMP9* gene polymorphisms are associated with nephrolithiasis[45]. *MMP9* is upregulated by classically activated macrophages during an inflammatory response and is also expressed by other inflammatory cells and osteoclasts[46–48]. MMP9 is proposed to play a role in renal fibrosis and epithelial-to-mesenchymal transition[49,50]. It also interacts with osteopontin to enhance macrophage chemotaxis and

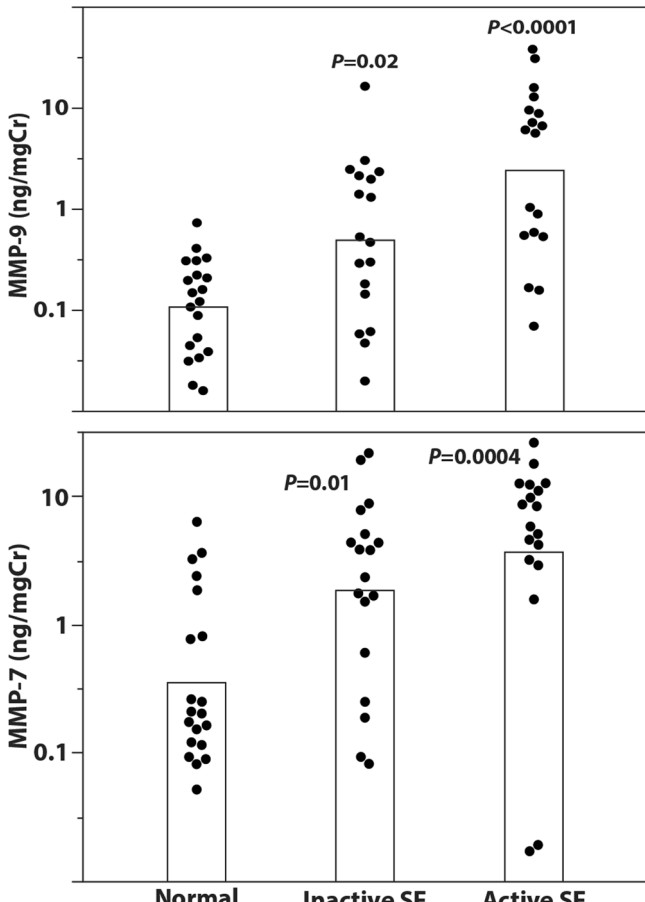

**Fig. 7 | MMP7 and MMP9 levels are increased in urine of CaOx stone patients and correlate with disease activity.** Urine samples were taken either from CaOx stone patients undergoing surgery for stone removal (active stone formers, SF, $N = 18$) or from patients who had previously been stone formers (Inactive SF, $N = 18$) or from healthy volunteers ($N = 20$). Demographics and relevant clinical variables for each group are presented in Supplementary Table 4. Samples were assayed for urine MMP7, MMP9, and urine creatinine (Cr) by ELISA according to the manufacturer's instructions. Samples are plotted as $\log_{10}$(ng MMP/mg Cr, top panel for MMP9 and bottom panel for MMP7). Statistical comparisons between groups were done using two-tailed ANOVA with the Tukey-Kramer post hoc test.

fibrosis[50]. More recently, experimental studies by Wu et. al. suggested that activation of ROS in kidney tubular epithelial cells by way of the NK-κB/MMP-9 pathway promotes crystal deposition in the kidney[51]. Therefore, upregulation of MMP9 could be an indicator of crystal deposition, immune activation, and transition towards a profibrotic phenotype.

Since our tissue work suggested a potential role for MMP7 and MMP9, we investigated whether measuring these molecules in the urine could be useful in patients with stone disease. Previous studies have investigated the utility of these markers in other forms of kidney disease[52–56]. To our knowledge, our study is the first to show that the urine levels of MMP7 and MMP9 are elevated in patients with the stone disease compared to non-stone-forming kidney subjects, particularly in the absence of a clinically detectable kidney function impairment. MMP7 and MMP9 urine levels are higher, particularly for MMP9, in patients with symptomatic active stone disease. Large studies are needed to define the utility of these markers in the management of patients with kidney stone disease. For example, longitudinal measurements of these markers and their changes may help predict stone recurrence and response to therapy, supplementing computed tomography to reduce lifetime radiation exposure in stone formers.

Indeed, our results provide a solid rationale to study these markers in the context of a large clinical trial.

Our study has limitations. Renal papilla samples from stone formers are challenging to obtain as they require a biopsy during a nephrolithotomy surgical procedure. Fortunately, our investigative team has substantial experience obtaining these samples[35,57]. The number of tissue specimens used to generate the transcriptomic and imaging datasets was relatively small, underscoring the importance and scarcity of these samples. Despite the sample size, the findings were validated by orthogonal methods, and the importance of the pathways and molecules uncovered were ultimately linked to urinary measurements in a larger clinical cohort. Further, the co-clustering of nuclei from the renal papilla with a publicly available snRNAseq whole kidney atlas (of more than 200,000 nuclei) allowed substantially more power to distinguish granular cell subtypes and cell states. Due to the cost-prohibitive nature of multi-omics studies, our approach of discovery in a small sample size and validation in a larger cohort serves as a viable model for hypothesis-driven future studies economizing on tissue and maximizing validation efforts on samples collected through non-invasive methods over time. Classifying cells at the protein level by the expression of few marker cells, particularly macrophages, may underrepresent the diverse phenotype of such cells. However, we used markers that are commonly available and consistent with available literature[26,30]. As discussed, the findings of elevated MMPs could also be observed in other forms of kidney disease, which could also be a limitation. However, the cohorts used here had normal kidney function, and the potential utility of these markers in tracking stone disease activity, particularly in patients with normal kidney function, is thereby not diminished. We studied CaOx kidney stone formers, which are the most common type of kidney stones. The applicability of our findings to other types of stone disease will need to be established by additional studies.

In conclusion, our integrated multi-omics investigation uncovered the complexity of the human kidney papilla and provides important insights into the role of immune system activation and mineral deposition. We established the diffuse injury associated with nephrolithiasis and highlight the importance of a matrix remodeling program in the kidney papilla. Particularly, we identified MMP7 and MMP9 as potential molecules that may serve as noninvasive markers of kidney stone disease course and activity. This investigation provides a strong foundation for future studies that explore the pathogenesis of kidney stone disease and the application of precision medicine to patients with nephrolithiasis.

## Methods
### Human samples sources
**Ethical compliance.** We have complied with all ethical regulations related to this study. All experiments on human samples followed all relevant guidelines and regulations. The relevant oversight information is given based on tissue sources (detailed below).

**Single nuclear RNA sequencing sample sources.** A combined snRNAseq atlas of 280,053 nuclei was derived from a total of 49 samples spanning both a new dataset (79,717 nuclei) and the combined Human BioMolecular Atlas Program (HubMap) (https://hubmapconsortium.org/hubmap-data/) and Kidney Precision Medicine Project (KPMP) (https://www.kpmp.org) datasets (200,336 nuclei), that is publicly available (GSE1183279[18]. Human samples collected as part of the KPMP consortium were obtained with informed consent and approved under a protocol by the KPMP single IRB of the University of Washington Institutional Review Board (IRB#20190213). Samples as part of the HuBMAP consortium were collected using informed consent by the Kidney Translational Research Center (KTRC) under a protocol approved by the Washington University Institutional Review Board (IRB #201102312). Single cells were isolated from frozen tissues as previously described[18].

Briefly, fresh frozen optimal cutting temperature (OCT)-embedded tissue was sectioned in 40 μm curls and 7–10 sections were pooled to isolate nuclei by mechanical dissociation. The quality was checked for morphology and processed using the 10x genomics pipeline in the Washington University core facilities. The HuBMAP / KPMP atlas contained data from 36 subjects and five subjects provided two samples each for a total of 41 samples. The presence of cortex, medulla, or papilla was defined by adjacent cross-sectional Periodic Acid Schiff-stained histological sections. In these 41 samples, there were 29 samples with cortex, 14 samples with medulla, and five samples with papilla. Seven subjects' samples contained both cortex and medulla. Of the five papilla samples, three were biopsy specimens from CaOx stone formers obtained at Indiana University (see below) and two were nephrectomy specimens (non-involved part of tumor nephrectomies with clear margins or deceased transplant donor) from non-stone formers obtained at Washington University. For this study, eight additional samples were added to expand the single nuclear RNA sequencing atlas, including 6 containing papillary tissue, 1 cortical sample, and 1 medullary sample. Of the 11 total (new samples and existing HuBMAP / KPMP) papillary samples, there were 76,126 papillary nuclei from 5 non-stone former samples (tumor nephrectomies or deceased donors), 5 CaOx stone former specimens, and one sample with only partial papilla from a deceased donor who had severe clinical acute kidney injury which was excluded from the analysis comparing reference and stone disease specimens. A summary of the tissue sources of the full papillary specimens, available clinical and demographic variables, and the distributions of the various assays performed are outlined in Supplementary Table 2.

**Human kidney papillae samples used in spatial technologies.** Patients with idiopathic CaOx stones underwent a renal papillary biopsy procedure during a clinically indicated elective percutaneous nephrolithotomy for stone removal[31], approved by the institutional review board (IRB) of Indiana University (IRB # 1010002261). Informed consent was obtained from all study participants. Human reference (non-stone formers) nephrectomy papillary specimens (donor nephrectomies not amenable to transplant because of time or anatomy without evidence of stone disease) were obtained from the Biopsy Biobank Cohort of Indiana[58] under a waiver of informed consent as approved by the Indiana University IRB (IRB #1906572234). Following extraction, papillary tissues were frozen on dry ice in OCT medium.

**Cohorts for urine MMP7 and MMP9 measurements.** Urine specimens from 20 healthy participants (Normal) with no personal or family history of kidney stones and 18 patients with a history of CaOx stones (Non-active stone formers) were obtained with informed consent from an ongoing study at the University of Chicago (IRB protocol 09-164B). Participants were studied in the General Clinical Research Center at the University of Chicago, and the urine specimens used in these studies were obtained during the same morning period. Urine samples from 18 patients with the active stone disease were obtained during elective percutaneous nephrolithotomy for the stone disease at Indiana University (IRB protocol 1010002261). The demographic and relevant clinical characteristics of these subjects are presented in Supplementary Table 4.

**snRNAseq and processing**
From an integrated HubMAP and KPMP atlas of renal cell types[18], the snRNAseq portion of the Seurat object, including papilla samples, was reproduced to identify cell type clusters. For quality control, 10X snRNAseq cell barcodes passing 10X Cell Ranger filters were used for downstream analyses. All mitochondrial transcripts were removed. Doublets were identified and removed with DoubletDetection software (v2.4.0)[59]. The additional samples added for this study were handled by the same lab (Jain Lab) that processed the HubMAP/KPMP snRNAseq atlas. A total of 49 samples were merged and only nuclei

barcodes with more than 400 and less than 7500 genes detected were maintained in the merged atlas. A gene unique molecular identifier (UMI) ratio filter was applied using Pagoda2 to remove low-quality nuclei (github.com/hms-dbmi/pagoda2)[60]. Expression was calculated in 10X Cell Ranger v3 after demultiplexing and barcode processing. The GRCh38 (hg38) reference genome was used for the snRNAseq and subsequent spatial transcriptomic datasets.

**snRNAseq clustering and annotation**
snRNAseq cluster definitions were adopted from the current version of the combined HuBMAP and KPMP atlas[18]. Briefly, nuclei were clustered with pagoda2. Total counts per nucleus were normalized, batch effect was corrected, and principal component analyses were performed using all significant variant genes ($N = 5526$). Initial cluster identities were determined by the infomap community detection algorithm. Using a primary cluster resolution of k = 100, all principal components and annotations were imported into Seurat v4.0.0 to create a merged uniform manifold approximation and projection (UMAP). Standardized anatomical and cell type nomenclature was used to annotate cell types and subtypes, based on the collaborative KPMP and HuBMAP definitions. These definitions were based on published datasets and the expertise of consortium pathologists, biologists, nephrologists, and ontologists[61–66]. Specifics of the cluster decision tree algorithm have been previously described[18]. Within the integrated HubMAP and KPMP atlas, putative adaptive and degenerative cell states were identified in epithelial sub-clusters with at least one of the following: reduced genes detected, higher mitochondrial transcript amount, higher ER associated transcript number, increased expression of known injury markers (e.g., IGFBP7, HAVCR1, LCN2, CST3, etc.), or enrichment in samples with acute or chronic kidney disease. Markers of these cell states were identified using the Seurat function "FindConservedMarkers" with the following parameter settings: grouping.var = "condition.l1", min.pct = 0.25, and max.cells.per.ident = 300. The gene set for the adaptive and degenerative cell states was trimmed to include only enriched genes at a $p$ value < 0.05 and mean log2 fold change > 0.6. Due to the consistent loss of epithelial cell type specific markers, all nuclei within the adaptive and degenerative cell state clusters were merged into a single undifferentiated epithelial cluster within the papilla snRNAseq UMAP.

**Visium Slide preparation, mRNA extraction, and sequencing**
Ten samples (4 references and 6 calcium oxalate) provided 10,695 spots for analysis. Frozen 10 μm sections were mounted onto etched frames of the Visium spatial gene expression (VSGE) slides according to 10x Genomics protocols (*Visium Spatial Protocols—Tissue Preparation Guide*, Document Number CD = G000240 Rev A, 10x Genomics). Tissue sections were fixed with methanol, subsequently stained with hematoxylin and eosin (H&E), and imaged by bright-field microscopy. Microscopic images were acquired according to protocols described in 10x Genomics protocols (Technical Note - *Visium Spatial Gene Expression Imaging Guidelines*, Document Number CG000241, 10x Genomics). H&E-stained sections were imaged with a Keyence BZ-X810 microscope equipped with a Nikon 10× CFI Plan Fluor objective at 0.7547 um/pixel and image resolution of 1920×1440. Images were collected as mosaics of 10x fields. Stained tissues were permeabilized for 12 minutes. mRNA bound to oligonucleotides on the capture areas of the Visium slides was extracted. cDNA libraries were prepared with second-strand synthesis and sequenced utilizing the NovaSeq 6000 Sequencing system (Illumina) in the 28 bp + 120 bp paired-end sequencing mode. Post-sequencing quality metrics are included in Supplementary Table 2.

**Spatial transcriptomics expression analysis**
Using Space Ranger 1.2.0 with the reference genome GRCh3-2020-A, samples were mapped, and counts were generated that corresponded

to the barcoded 55 μm spot coordinates within each fiducial frame. This allowed the association of read counts with their location within the H&E image. Space Ranger calculated differential expression between assigned clusters using sSeq and edgeR. The data were normalized by SCTransform and merged to build a unified UMAP and dataset using Seurat. The normalization was corrected for batch effect[67]. All feature plots show expression after normalization.

## Selection of histologic phenotypes

Kidney stone papillary tissue biopsies were manually annotated utilizing the 10x Genomics Loupe Browser (10x Genomics Loupe Browser 5.0.0). Specific regions of the stone tissue papilla were categorized as either non-mineralized, contiguous to mineral, or mineralized regions. Spots directly overlying evident mineral precipitation were designated as mineralized; spots falling within a three-spot radius of the annotated mineral precipitation were categorized as "contiguous" to mineral; all remaining spots were designated as non-mineralized. Mineral deposition spots were validated by James Williams who is an expert in papillary stone histology[25].

## Pathways and gene expression analysis

Differential expression comparisons across samples were performed using R and the R packages ReactomePA and ClusterProfiler as previously described by Ferreira et al.[23]. Briefly, the DEGs in each comparison were found with the Seurat function FindMarkers and tested with a Wilcoxon's rank sum test. Pathway enrichment for those genes was performed with the R packages ReactomePA[68] and ClusterProfiler[69]. Statistical significance is calculated based on a hypergeometric distribution[68,69], and Bonferroni-adjusted $p$-values are presented in volcano and pathway plots. A Wilcoxon rank sum test was used to compare the proportion of undifferentiated cell states across samples. For distribution bar plots, the proportion of cells was compared by Fisher's exact test and the p-values are unadjusted with the level of significance clearly denoted.

## Label transfer

Seurat v3.2 was used to accomplish label transfer from snRNA-Seq cell types and cell states to spatial transcriptomic spots in each sample. For deconvolution analyses, a Seurat v3.2 anchor methodology was used to transfer single-cell cluster information to Visium data as described[23]. Each spot receives a probability or transfer score for its association with a given snRNA-Seq cell type or cell state cluster. The transfer scores are summed, and each spot is deconvoluted with the fractions of the spot corresponding to the relative proportion of transfer score of each contributing snRNAseq cluster. A pie chart is displayed over the spatial transcriptomic sample image.

## Single molecular Fluorescence in situ hybridization (smFISH)

To determine the cell-specific gene expression patterns and to validate the differentially expressed genes, we performed RNAscope-based smFISH approach as described previously[70–72]. Briefly, smFISH was performed using RNAscope Multiplex Fluorescent V2 kit (Advanced Cell Diagnostics, Newark, CA) in formalin-fixed and paraffin-embedded (FFPE) human kidney cross-sections (5 μm) from patients with Calcium oxalate kidney stone disease. Slides were deparaffinized, followed by pretreatment with hydrogen peroxide for 10 min at room temperature (RT then antigen retrieval was performed for 15 min at 99 °C, as per the manufacturer's protocol. Then the slides were hybridized with probes for *AQP2, SLC26A7, MMP7, PROM1*, and *IGFBP7* mRNA and the signals were developed using secondary TSA plus fluorophores l (Perkin Elmer, Waltham, MA). The slides were counterstained with DAPI to detect the nucleus and coverslips were mounted using ProLong Gold antifade mounting media (Thermo Fisher, Rockford, IL). Images were acquired using an LSM800 confocal microscope (Carl Zeiss, Germany). To determine the difference in fluorescence intensity of cells within

collecting ducts, we used ImageJ to manually segment cells and measure the mean fluorescence intensity (Fig. 1p and Supplementary Fig. 4a).

## Tissue processing, immunofluorescence staining and large-scale 3d confocal imaging

Papillary biopsies were immediately immersed in OCT medium and frozen on dry ice. For immunofluorescence analysis, tissues were cryosectioned (Leica Biosystems, Wetzlar, Germany) at 20 μm thick sections. Sections were washed in phosphate-buffered saline (PBS), and fixed for 4 h at room temperature (RT) in 4% paraformaldehyde. Next, the sections were washed in PBS two more times and then blocked in 10% normal donkey serum for two hours at RT. Primary antibodies were added in blocking buffer and tissues were incubated at RT overnight. When targeting intracellular antigens, permeabilization was performed using 0.2% Triton X (Santa Cruz Biotechnology, Inc., Dallas, TX) (Gildea, 2017)[73]. The following primary antibodies were used for detection: anti-aquaporin 1 (Santa Cruz Biotechnology, Inc., Dallas, TX; sc-32737, dilution 1:50), anti-CD68 (Agilent Technologies, Santa Clara, CA; M087601, dilution 1:50), anti-phospho-c-JUN (Cell Signaling Technology, Danvers, MA; 9261, dilution 1:50), anti-aquaporin 2 (Life Technologies, Carlsbad, CA; PA522865, dilution 1:50) and anti-MMP7 (Santa Cruz Biotechnology, Dallas, TX; sc-515703, dilution 1:25). After washing with PBS, the following Alexa Fluor (ThermoFisher Scientific, Waltham, MA) dye-conjugated secondary antibodies were added: donkey anti-mouse-488, anti-mouse-568 and anti-rabbit-647. 4′,6-Diamidino-2-phenylindole (DAPI) (Abcam, Cambridge, United Kingdom; ab228549) was used for staining nuclei. Subsequently, sections were washed three times for 30 minutes each in PBS and then fixed in 4% paraformaldehyde for an additional 15 minutes. After a final wash in PBS for 30 minutes, sections were mounted on a glass slide using ProLongTM Glass Antifade Mountant, (Thermo Fisher Scientific, Waltham, MA; REF# P36980). Images were sequentially acquired in 3 or 4 separate channels using the Leica SP8 confocal microscope and collecting whole volume stacks using 20xNA 0.75 objectives with 1.0 μm spacing. Mosaics were stitched using Leica LAS X software to generate large-scale 3D images. A negative control without primary antibodies was used to ensure the absence of non-specific binding of secondary antibodies. Microscope settings were identical among imaging sessions for each specimen when imaging the same probes.

## 3D tissue cytometry

3D tissue cytometry was performed on image volumes using the volumetric tissue exploration and analysis (VTEA) software[74,75]. Segmentation settings were adjusted to yield the best result which was verified visually by sampling random fields within each image stack. Fluorescence from phospho-c-Jun and CD68 was associated with nuclei by 3D morphology and displayed on a scatterplot as individual points, allowing gating of specific cell populations based on fluorescence intensities. Results were reported as the percentage of total cells segmented in each large-scale image volume[15]. For MMP7 analysis, unsupervised analysis was performed after segmentation, using MMP7 and AQP2 fluorescence intensity. Cell clusters were identified using G-means clustering and dimensionality reduction visualization using $t$-distributed stochastic neighborhood embedding ($t$-SNE). The identity of the cell clusters was verified by plotting their mean intensities for specific makers using violin plots and heatmaps for visualization, and directly mapping the cell clusters on the image volumes using nuclear overlays This analytical pipeline is described in a recent publication[75]. The efficacy of detecting MMP7+ collecting duct cells was done using the manual image-based labeling function of VTEA. Forty to fifty cells were randomly selected from each tissue and labeled by a renal expert to generate a ground truth which was compared to the unsupervised classification strategies. The sensitivity and specificity of detecting MMP7+ collecting duct cells were 89% and 81%, respectively.

## CODEX antibody conjugation and validation

A total of 32 antibodies were used for CODEX (Supplementary Table 3) and 19 of them were conjugated in-house using a protocol outlined by Akoya Biosciences (Akoya Biosciences, Inc. CODEX® User Manual, Menlo Park, CA). Conjugation of antibodies to their assigned barcodes, they were first reduced using a "Reduction Master Mix" (Akoya Biosciences, Menlo Park, CA). Lyophilized barcodes were then resuspended using the Molecular Biology Grade Water and Conjugation Solution. The barcode solution was then added to the appropriate reduced antibody and incubated for two hours at RT. After incubation, the newly conjugated antibody-barcode was purified in a three-step wash/spin process and stored at 4 °C. Successful conjugation was validated via gel electrophoresis as well as immunofluorescent staining and confocal imaging (Supplementary Fig. 6).

## CODEX imaging

Human renal tissue sections of 10 μm were cut from OCT blocks onto poly-L-lysine coated coverslips. Sections were prepared as detailed by the manufacturer's instructions (Akoya Biosciences, Menlo Park, CA). Tissue retrieval was conducted with a three-step hydration process, followed by a PFA fixation. During fixation, an antibody cocktail of the 32 antibodies listed in Supplementary Table 3 was made and then dispensed onto the coverslip. Tissues were allowed to incubate with the staining solution overnight at 4 °C. The following day, the staining solution was washed from the tissues, and a multi-step fixation occurred[76].

The imaging of tissues was conducted at 20x resolution using the CODEX system from Akoya Biosciences and a Keyence BZ-X810 slide scanning microscope (Keyence Corporation, Itasca, IL). The resulting images were processed using the CODEX processing software (Akoya Biosciences, Menlo Park, CA) and visualized using FIJI/ImageJ.

## CODEX quality control

Major components of quality control checks for this assay pertain to tissue quality, reagent quality/antibody validation, instrument performance, cyclical image acquisition, and image analysis. Tissue quality: verified by contiguous section with histological staining. For an antibody to pass validation, the pattern of the conjugated antibody in the post-conjugation staining and in the CODEX run should produce a similar pattern to that of the unconjugated antibody (Supplementary Fig. 6). For antibodies with a known target and expected staining of cell pattern, we expect to see labeling that is consistent with the pattern described in the literature and other experimental studies. For injury and cell state markers, positive controls are less well defined, and we rely on the fact that the signal is limited to specific cells, around morphologically altered areas with evidence of injury, inflammation or fibrosis.

Instrument performance quality is performed based on the manufacturer's instructions. For cyclical image collection, the specificity of target detection depends critically upon the efficiency with which previously introduced probes are stripped from the sample prior to imaging. The efficiency of stripping is evaluated by comparing each image with the next image collected for that same channel (Supplementary Fig. 7). This is also evident in the pattern of staining where the mean intensity fluorescence for any potential target with carryover is evaluated for each cell class (Supplementary Fig. 8). Carryover, if present, would be apparent in the appearance of artifactual objects with characteristic combinations of probes from the sequence of one channel and these objects can be specifically removed from the analysis. Quantitation of probe fluorescence depends upon the elimination of background fluorescence, which is identified in images of unlabeled tissues collected prior to the first round of imaging of each channel. The AKOYA software eliminates background autofluorescence by subtracting pixel-for-pixel values of these "blank" images from each of the images of labeled tissue.

## Unsupervised analysis, clustering, and mapping of cell types

CODEX images were segmented and analyzed using VTEA version 1.0.3[75]. Cell clusters were identified using unsupervised Ward clustering and dimensionality reduction visualization using t-distributed stochastic neighborhood embedding (t-SNE). The identity of the cell clusters was verified by plotting their mean intensities for specific makers (Supplementary Table 3) using violin plots and heatmaps for visualization, and directly mapping the cell clusters on the image volumes using nuclear overlays (Fig. 2). Based on this schema, epithelial cells (e.g. collecting duct and thin limbs, with appropriate morphology by cell mapping and high expression of epithelial cell markers and lacking immune and endothelial cells), immune cells (mapping to interstitium and high expression of immune cell markers, the subtypes of which is based on cell type-specific marker, e.g. T cells are CD3 + ), endothelial cells (mapping to interstitium and positive for CD31) and stromal cells (mapping to interstitium and high level of expression of fibroblast or myofibroblast markers) were identified (Fig. 2, Supplementary Figs. 7 and 8). The sensitivity and specificity of CODEX analysis were determined for the detection of collecting duct cells in the specimen used as described for 3D tissue cytometry (above), and were 100% and 87%, respectively.

## ELISA and urine collection

Urine samples were collected from patients with a history of CaOx stone formation (non-active stone formers), from CaOx stone patients immediately before undergoing surgery for stone removal (active stone formers) or from non-stone formers as described above. The urine samples were immediately frozen at −80 °C. At the time of analysis, samples were thawed and cleared by centrifugation for 5 minutes at 1000xg and assayed for urine MMP7 and MMP9 by ELISA according to manufacturer's instructions (R&D Systems, DMP700/DMP900, Minneapolis, MN). Urine creatinine was used to normalize MMP7 and MMP9 levels between non-active, active, and non-stone formers.

## Statistics and reproducibility

Statistics used within transcriptomics analyses are described above using R. For the imaging data, unless otherwise specified, statistical analysis and graphing were done using GraphPad Prism. For each figure, the statistical test used is specified in the text of the figure legend. Generally, for normally distributed samples, a student's t-test was used to compare the means, and an F-test was performed to ascertain the equality of variances. Urine levels of MMP7 and MMP9 were compared using ANOVA with the Tukey-Kramer post hoc test using JMP software. Because of the nature of the experiments and the rarity of the specimens involved, all the experimental data is presented in the figures, supplementary information, and source data. Replicates, when shown, are biological. For each specimen, we specified in Supplemental Table 2, in which figure it was used.

## Reporting summary

Further information on research design is available in the Nature Portfolio Reporting Summary linked to this article.

# Data availability

All sequencing data were analyzed using the GRCh38 (hg38) reference genome. For published data from the KPMP/HuBMAP atlas, access to the raw snRNAseq data has been detailed in the Lake et al manuscript (Nature 2023, in press, https://doi.org/10.1038/s41586-023-05769-3). These raw sequencing data are under controlled access (human data) as they are potentially identifiable and can be accessed from the following respective sources: 1) KPMP data has been deposited at https://atlas.kpmp.org/repository/ and can be requested and made available by signing a data use agreement (DUA) with KPMP. KPMP will respond to initial data requests within 12–36 h and provide data up to one month after DUA has been signed.

Manuscripts resulting from KPMP controlled access data are requested to go through the KPMP Publications and Presentations Committee which reviews and approves manuscripts every two weeks. Any analysis resulting from KPMP data may be published or shared so long as it does not re-identify KPMP participants. 2) The HuBMAP raw data are available for download from the database of Genotypes and Phenotypes (dbGaP, phs002249) by requesting for authorized access following instructions on the dbGaP website. The process to request access to this dbGaP study is here: https://dbgap. ncbi.nlm.nih.gov/aa/wga.cgi?adddataset=phs002249&page=login. Briefly, to download the human sequence data for this study after obtaining authorization from the NIH Data Access Committee, one would go through the Sequence Read Archive here: https://www. ncbi.nlm.nih.gov/bioproject/PRJNA671343. Processed data (published in the Lake et al manuscript) for both of these sources is available in the GEO Super-series: GSE183279 (https://www.ncbi.nlm. nih.gov/geo/query/acc.cgi?acc=GSE183279). The new snRNAseq data generated for this study (raw sequencing and processed), along with all the Visium spatial transcriptomic data are available in GEO as the Super-series GSE231630 (https://www.ncbi.nlm.nih.gov/geo/query/ acc.cgi?acc=GSE231630 or deposited in the HuBMAP portal (https:// portal.hubmapconsortium.org/). A breakdown of the data availability of all papillary samples is presented in Supplementary Table 2. Source data are provided with this paper. For the imaging data: the source data for all the main and supplementary figures (3D imaging, CODEX, and smFISH) are available at the following repository: https://doi.org/10.5281/zenodo.7653239 and https://zenodo.org/ record/7653239#.ZGIuGc7MKUk Source data are provided with this paper.

## Code availability

The R packages and code used in the snRNAseq and spatial transcriptomics analysis are described in the methods. For image analysis, the VTEA software is available for download through the FIJI plugin updater with the source code is available on GitHub, https://github.com/icbm-iupui/volumetric-tissue-exploration-analysis. Additional descriptions and vignettes demonstrating the use of VTEA are available at https://vtea.wiki.

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

## Acknowledgements

The authors would like to thank Sharon Bledsoe, Jennifer Stashevsky, and Katrina Kelly for help with tissue specimens. We also thank KPMP participants and the HubMAP consortium. This work was funded from the following grant support from the National Institute of Health: P01 DK056788 (E.W, J. C.W., J.E.L., T.M.E-A), U01 DK114923 (T.M. E-A, M.E.), P30 DK079312 (T.M.E-A), R01 DK111651 (T. M. E-A), U54 DK134301 (S.J, T.M.E-A).

## Author contributions

Coordination of manuscript writing and project: V.H.C., W.S.B., M.T.E., J.C.W., T.M.E. Contribution to Patient Recruitment and Tissue Collection: S.J., F.L.C., E.W., J.C.W., J.E.L., Contribution to Tissue Processing: W.S.B., S.J., J.P.G., J.C.W., Y.-H.C., T.M.E. Contribution to Imaging data generation: W.S.B., A.R.S., F.S., D.B.,M.F., T.M.E., Contribution to snRNAseq generation: S.J., B.B.L., K.Z. Contribution to ST data generation: R.M.F., Y-H.C., M.T.E., Contribution to image data analysis: W.S.B., A.R.S., S.W., K.A.L., T.M.E., Contribution to transcriptomics analysis V.H.C., W.S.B., R.M.F., M.E., K.A.L., B.B.L., S.J., T.M.E., contribution to data interpretation: all authors, Contribution to writing manuscript: all authors.

## Competing interests

The authors declare no competing interests.

## Additional information

[1]Department of Anatomy, Cell Biology and Physiology, Indiana University School of Medicine, Indianapolis, IN, USA. [2]Department of Medicine, Division of Nephrology, Indiana University School of Medicine, Indianapolis, IN, USA. [3]Center for Diabetes and Metabolic Diseases, Herman B. Wells Center for Pediatric Research, Indiana University School of Medicine, Indianapolis, IN, USA. [4]Department of Urology, Indiana University School of Medicine, Indianapolis, IN, USA. [5]Department of Pathology and Microbiology, University of Nebraska Medical Center, Omaha, NE, USA. [6]San Diego Institute of Science, Altos Labs, San Diego, CA, USA. [7]Department of Pathology and Immunology, Washington University, St. Louis, MO, USA. [8]Department of Medicine, Division of Nephrology, University of Chicago, Chicago, IL, USA. [9]Department of Medicine, Division of Nephrology, Washington University, St. Louis, MO, USA. [10]Department of Medical and Molecular Genetics, Indiana University School of Medicine, Indianapolis, IN, USA. [11]Department of Medicine, Indianapolis VA Medical Center, Indianapolis, IN, USA. [12]These authors contributed equally: Victor Hugo Canela, William S. Bowen, Ricardo Melo Ferreira. ✉e-mail: sanjayjain@wustl.edu; meadon@iupui.edu; jwillia3@iu.edu; telachka@iu.edu

## the Kidney Precision Medicine Project

**Ricardo Melo Ferreira** [2,12], **Daria Barwinska** [2], **Seth Winfree** [1,5], **Blue B. Lake** [6], **Joseph P. Gaut**[7], **Michael Ferkowicz**[2], **Kun Zhang** [6], **Sanjay Jain**[9]✉, **Michael T. Eadon** [2,10]✉, **James C. Williams Jr.** [1]✉ & **Tarek M. El-Achkar** [1,2,11]✉

A full list of members and their affiliations appears in the Supplementary Information.

