## [Peer Review File · Nature Communications]

A spatially anchored transcriptomic atlas of the human kidney papilla identifies significant immune injury in patients with stone diseaseREVIEWER COMMENTS

Reviewer #1 (Remarks to the Author):

In this manuscript, Canela et al. characterized the renal papilla of 2 human normal reference individuals (controls) and 3 stone carriers using snRNA-seq, spatial transcriptomics and CODEX imaging (using a panel of 32 marker proteins). snRNA-seq data were part of publicly available datasets from the Human BioMolecular Atlas Program (HuBMap) (<https://hubmapconsortium.org/hubmap-data/>) and Kidney Precision Medicine Project (KPMP) (<https://www.kpmp.org>) datasets. They also used additional samples for 3D tissue cytometry based on phospho-cJUN and CD68 stainings. In addition, they analyzed urines from healthy individuals (N=20), inactive stone formers (N=18) and active stone formers (N=18) for MMP7 and MMP9. They draw the following key conclusions:

1. "...we defined subtypes of immune, stromal and principal cells enriched in the papilla, and characterized an undifferentiated epithelial cell cluster that was more prevalent in stone patients". The authors propose that the "undifferentiated" cell cluster "that this cluster may represent a final common injury phenotype derived from epithelial cells with multiple origins"
2. "The microenvironment of mineral deposition had features of an immune synapse with antigen presenting inflammatory macrophages interacting with T cells"
3. "The expression of MMP7 and MMP9 was associated with stone disease and mineral deposition, respectively. MMP7 and MMP9 were significantly increased in the urine of patients with CaOx stone disease compared to non-stone formers, and their levels correlated with disease activity in stone formers."

Strengths:

1. Combination of three high-throughput approaches, including two transcriptomics-based approaches (snRNA-seq, spatial transcriptomics) and one protein based approach (CODEX) that provides potentially new insights into the biology of the human kidney papilla and into the pathophysiology of renal stone disease.
2. Detailed description of the cell type and cell state composition of the human kidney papilla in stone disease patients and controls.

Weaknesses:

1. Each of the three conclusions above is rather preliminary (see below).
2. Small number of individuals studied (snRNA-seq: N=3 stone patients, N=2 controls; Spatial transcriptomics: N=2 stone patients, N=2 controls; CODEX: N=1 stone patient, N=1 control).
3. Data integration and presentation need improvement.

Major concerns:

- The identification of novel injury cell states associated with stone disease constitutes a key finding of this paper (see conclusion 1). However, it needs to be clearly shown how PC1, PC2 and "undifferentiated" cells are distributed among stone carriers and controls, including quantitative statistics on a patient level.
- The authors claim that cell populations uncovered by snRNA-seq and ST (PC1, PC2, undiff) correspond to cell populations detected by CODEX (CD-1, CD-2, undiff) and claim that this serves as validation of the existence of these three subtypes. However, this conclusion is based on a few highly selected markers (CDH1 for PC1/2 and IGFBP7/PROM1 for undifferentiated cells), whereas other markers do not seem to support this conclusion (SPP1, LC3). The authors should use multiplex in situ hybridization (e. g. RNA-Scope) to validate the existence of PC1, PC2 and undifferentiated cells using ideal markers derived from their transcriptomics studies.
- I do not agree with conclusion 2 above, since evidence of "immune synapses" is lacking. It would be better to downstate this conclusion and say precisely what new knowledge is generated by this study. The interaction of mineral plaques with immune cells is - of course- well known.
- Conclusion 3 above: MMP7 and MMP9 were identified as urine markers of stone disease. As mentioned in the discussion section, these markers have been previously associated with other

kidney diseases and may be indicative of injury rather than being specific to stone disease. This limitation should be clearly stated.

- Cell type annotation is very superficial (Fig. 1A, B). The authors need to present marker gene expression per cluster identified in Fig. 1A and Fig. 1B. Some cell type abbreviations are not even defined (e. g. "NEU" cells). A comparison with the published literature is lacking.
- Differential expression in cell types of the kidney, including principal cells across the corticomedullary axis has been studied extensively, including single cell approaches in mice and humans. Again, the results obtained in this study need to be compared to the known literature and literature needs to be cited appropriately.
- Corticomedullary differences in gene expression, e. g. of ENaC subunits (cortical enriched) or urea transporter UT2 (medullary enriched) need to be compared vs. the published literature. Why is it important that RALYL is higher in cortex than in medulla (results section)? The authors claim that distinct bicarbonate transporters are expressed in collecting duct cells, but they do not say which transporters they refer to.
- It is surprising to see that AQP2 is expressed in intercalated cells. This raises further concerns about cell type annotation (to my knowledge AQP2 is strictly specific to PC).
- Data need to be made available to the community beyond this manuscript, e. g. via web interface or public databases.
- The title of the manuscript ("A spatially anchored transcriptomic atlas of the human kidney papilla identifies significant immune injury and matrix remodeling in patients with stone disease") is somewhat misleading, since no evidence of matrix remodeling is presented (only differential expression of MMPs).

Minor points:

- Abbreviations are used inconsistently. E. g. beta-catenin is called "b-catenin" in Suppl. Table 3, but CTNNB1 in Fig. 1U. Osteopontin is referred to as OPN, SSP1 (Supp. Table 3) and SPP1 (Main text and Fig. 1Q/W). This makes the paper very hard to read.
- Fig. 1: It should be clearly stated, which of the panels refer to snRNA-seq, ST and CODEX.
- Fig. 1W: The authors claim to have identified interstitial cells in the papilla and to have divided them into separate cell classes based on unsupervised analyses. However, no data are shown to support this. Fig. 1W simply shows violin plots of 6 selected markers.
- Fig. 2F: For some cell types, no data appear to be available in stone carriers (IC, TAL). Please explain.
- Fig. 6: It is unclear how Fig. 6 connects with the rest of the paper. The authors show phospho-cJUN-positive cells and claim a connection to ROS. What is the identity of these cells? Why was this done and how does it support the conclusions of this study?
- Fig. 7: Please indicate N per group and statistical tests used in figure legend.
- Fig. 1Q: *** denotes $p < 0.01$ ◊ Which statistical test was used?

Reviewer #2 (Remarks to the Author):

Overview

The authors present a spatially anchored transcriptomic analysis of human renal papillary biopsy samples obtained from stone formers using integrated snRNA seq and spatial transcriptomics, along with large scale multiplexed 3D IF, and 2D CODEX imaging to identify the spatial localization of specific cell types in the papilla, and changes that occur in stone formers. This is an unprecedented look into the cell biology of the human renal papilla using rare tissue samples obtained from ureteroscopic renal papillary biopsies obtained surgically from calcium oxalate stone forming patients.

They identify subpopulations of collecting duct cells and undifferentiated epithelial cells that are enriched in stone formers. They also show that sites of mineral deposition are active immune zones with features of immune injury and matrix remodeling genes, including MMP7 and MMP9, that extend beyond the areas of mineralization, and provide orthogonal validation for the MMP7 and 9 data showing in an independent cohort of patient derived urine samples that MMP7 and 9 levels are increased in stone forming patients.

The problem with the paper, as acknowledged by the authors, is the small number of samples included in this analysis. This is obfuscated over the course of the manuscript in a number of ways: 1) while snRNA seq data was derived from a much larger KPMP/HuBMAP kidney database, this only included 5 papillary samples, 3 from stone formers and 2 controls; 2) sub-analysis of papillary cell types was performed for most of the remaining studies and since these sub-populations of papillary cells are now a sub-population of a sub-population of cells, snRNAseq cell numbers are even rarer; and 3) this included only since even smaller numbers are used for some orthologous assays included in the study that are used to confirm snRNAseq and spatial transcriptomics data and used to define subpopulations of cells in the samples (e.g.; 1 control and 1 stone former included in CODEX analyses was used to characterize subpopulations of cells seen in stone formers vs. controls). The one orthologous assay that was used to study the largest number of samples, 3D IF, which was performed 4 controls and 4 stone formers, was limited to 2 readouts, CD68, which is a pan monocyte/macrophage marker, and p-cJUN, which is a non-specific marker of cellular stress, and confirmed, not unexpectedly, that regions of mineralization in stone formers contain inflammatory cells and there is cellular stress.

As a reviewer, I appreciate how hard it is to obtain these rare patient samples. On the other hand, the conclusions being made by the authors based on these small numbers of samples are not really supported by the data. Additional samples need to be used for validation, using appropriate orthologous methods, including in situ hybridization and/or more targeted immunofluorescence studies to draw these conclusions. In addition, much more information needs to be provided about validation and cell type specificity of antibodies used in the CODEX studies, and criteria used based on these markers to define subpopulations of cells (epithelial and macrophages) since it is not apparent in the manuscript.

Other/detailed concerns

- 1) The authors should also provide information about the reason for the nephrectomies, and whether papilla samples were tumor free, and the clinical indication for ureteroscopy (for example whether obstructed)
- 2) Histological analysis needs to clarify how and who identified mineralized regions of the samples
- 3) Overlay with histology. Are these the same samples or sequential sections? If these are sequential sections, how were the images registered to allow for differences between sections?
- 4) CODEX but no details provided about whether AB localization was compared with other antibodies raised against the same protein, QC validation sections etc., validation after different numbers of CODEX cycles etc.
- 5) Provide glossary of abbreviations used in figure legends
- 6) Histology/ST overlays in F/I/L and O are almost impossible to see and need to be accentuated. Better would be to overlay with MxIF/CODEX to properly identify and validate results.
- 7) Much more information needs to be provided to explain how the anticipated cell types were identified by CODEX using 32 markers, many of which are non-specific (e.g.: Krt8, which is touted as CD specific (Figure 1 U) but in fact is pan-epithelial, and while it strongly co-localizes with AQP2, also co-stains with AQP1, and LRP2, LTL, Uromodulin, SLC12A3 and Calbindin). The authors also need to comment on markers used to define macrophage subpopulations, since from list in S Table 3, the numbers of markers are small (CD68, CD206, and perhaps HLA-DR). Moreover, the authors state in S Figure 3 and in the text that CD68 is a marker of "activated" and "inflammatory" macrophages. This is incorrect; CD68 is pan-monocyte/macrophage marker
- 8) Color coding for a number of the multiplex images is not clear. Also staining with Thy1, which is not included in the list of CODEX Abs in S Table 3. Likewise no color apparent in Fig 1X
- 9) ST validation of MMP7 in CD lacks resolution and ability to differentiate CD from non-CD structures: a) histology provided does not differentiate CD from other epithelium (ideally this should be by IF); b) ST lacks resolution to ID cell types from these overlays anyway. Some form of multiplexed in situ hybridization in other samples, not used for ST analysis, would be appropriate here
- 10) Likewise they use ST in mineralized vs. non mineralized and contiguous regions showing increase in certain genes in contiguous regions, focusing on MMP9, CHIT1 and SSP1 by ST. This is appropriate for ST spatial resolution but should also be validated on additional samples using an

orthologous method (e.g.: ISH).

Reviewer #3 (Remarks to the Author):

In this manuscript the authors described the cellular and molecular mechanism involved in human renal papilla in patients with calcium oxalate stone disease compared to healthy subjects. This is a great effort to understand the mechanism of kidney stone specifically the contribution of immune system and to explore whether this underlying mechanism is generalized or limited to the cells involved with mineral deposition in the renal papilla. I have a few comments. 1) I understand one of the limitations of this study was difficulty in obtaining human kidney papilla biopsy specimens. Would the author explain in how many patients and normal posses enough samples to do spatial transcriptomic study and single nuclear RNA sequencing. 2) would the authors explain the specificity and sensitivity of high resolution large-scale multiplexed 3D and Co-Detection by indexing (CODEX). 3) As authors clearly indicated, the cellular mechanisms described elegantly are global and as they alluded to, such as global event would not provide sufficient knowledge toward limited cell niches associated with mineral deposition in the renal papilla. 4) Although the manuscript is very well written, it is extremely heavy in text, specifically in the discussion section. I highly recommended specific attention to be made in summarizing the discussion.

We thank the reviewers for the feedback received for our manuscript. We carefully took into account all the suggestions, and made significant changes to the manuscripts, which required many new experiments. We ran several additional samples (more than doubled) through single nuclear RNA sequencing, spatial transcriptomics, in-situ hybridization, and additional fluorescence imaging. The expanded data solidified our initial findings and clarified the points raised by the reviewers. Below is the point-by-point response:

Reviewer 1:

General response: Thank you for the thorough review and the encouraging comments about the strength of the study. We have done additional experiments on more samples to strengthen the results.

Weaknesses:

1. Each of the three conclusions above is rather preliminary.

Response: We have expanded our studies as suggested to solidify conclusions

2. Small number of individuals studied (snRNA-seq: N=3 stone patients, N=2 controls; Spatial transcriptomics: N=2 stone patients, N=2 controls; CODEX: N=1 stone patient, N=1 control).

Response: We have increased the sample sizes for most technologies: snRNAseq (N=5 stone patients, N= 5 controls), Spatial transcriptomics (N=6 stone patients, N=4 controls), Single molecule fluorescence in situ hybridization-FISH (N=4 stone patients), Additional 3D imaging (N= 3 stone patients, N= 3 controls, on top of the N=4 stone and N=4 controls for the first round of imaging). Despite the rarity of these samples and the associated efforts from various technologies, the data presented has been expanded and would be likely the best representation feasible for papillary biology and stone disease.

3. Data integration and presentation need improvement.

Response: We have implemented all the suggestions from the referees to improve data integration and presentation. We have separated each technology in presentation of the data to make it easier to follow.

Major concerns:

- The identification of novel injury cell states associated with stone disease constitutes a key finding of this paper (see conclusion 1). However, it needs to be clearly shown how PC1, PC2 and “undifferentiated” cells are distributed among stone carriers and controls, including quantitative statistics on a patient level.

Response: This point is well taken. In addition to reporting the differences in distribution of cell types between groups, we now included data to reflect patient level changes in transcriptomics (**Figure 3B**) and at the cellular protein level (**Supplementary Figure 9** and **Figure 3K**). The new data corroborates the increase in undifferentiated cells with stone disease at the patient level. Our data also suggest that stone disease induced a shift of collecting ducts cells towards the injured PapPC2 cell state (although this quantitative shift more evident with imaging than with snRNAseq). The increase in injury and inflammation is also determined on a patient level in the quantitative 3D imaging (**Figure 6**).

- The variable and that cell populations uncovered by snRNA-seq and ST (PC1, PC2, undiff) correspond to cell populations detected by CODEX (CD-1, CD-2, undiff) and claim that this serves as validation of the existence of these three subtypes. However, this conclusion is based on a few highly selected markers (CDH1 for PC1/2 and IGFBP7/PROM1 for undifferentiated cells), whereas other markers do not seem to support this conclusion (SPP1, LC3). The authors should use multiplex in situ hybridization (e. g. RNA-Scope) to validate the existence of PC1, PC2 and undifferentiated cells using ideal markers derived from their transcriptomics studies.

Response: Thank you for this comment. We now performed new experiments with in situ hybridization and additional targeted IF studies that show the existence of PapPC1, PapPC2 and undifferentiated cells using conventional and overlapping markers from the transcriptomics data. We also expanded the CODEX data to comprehensively characterize the expression profile of these cell types and highlight the differences in various cell markers that overlap with the transcriptomics data.

I do not agree with conclusion 2 above, since evidence of “immune synapses” is lacking. It would be better to downstate this conclusion and say precisely what new knowledge is generated by this study. The interaction of mineral plaques with immune cells is - of course- well known.

Response: This conclusion was down stated, and we removed the reference to immune synapse

Conclusion 3 above: MMP7 and MMP9 were identified as urine markers of stone disease. As mentioned in the discussion section, these markers have been previously associated with other kidney diseases and may be indicative of injury rather than being specific to stone disease. This limitation should be clearly stated.

Response: This is now further emphasized in the discussion as suggested.

Cell type annotation is very superficial (Fig. 1A, B). The authors need to present marker gene expression per cluster identified in Fig. 1A and Fig. 1B. Some cell type abbreviations are not even defined (e. g. “NEU” cells). A comparison with the published literature is lacking.

Response: The detailed cell annotation for the general cell atlas (**Figure 1A**) has been presented in detail in another publication (Lake et al ¹). For the papilla, we now present a new dot plot to detail the marker gene expression for each cell type in the papilla (**Figure 1E**). We also included a new **Supplementary Table 1** that clarifies all the abbreviations used. We also expanded characterizations of papillary cell types of interests, as suggested by the reviewers. Published data on human kidney papilla at the single cell level is limited, but we included more extensive literature on what is known from the collecting duct and other relevant cell types from animal studies.

Differential expression in cell types of the kidney, including principal cells across the corticomedullary axis has been studied extensively, including single cell approaches in mice and humans. Again, the results obtained in this study need to be compared to the known literature and literature needs to be cited appropriately.

Response: Comparison to the known literature of cell types on interest in the corticomedullary is now expanded as suggested by reviewer.

Corticomedullary differences in gene expression, e. g. of ENaC subunits (cortical enriched) or urea transporter UT2 (medullary enriched) need to be compared vs. the published literature. Why is it important that RALYL is higher in cortex than in medulla (results section)? The authors claim that distinct bicarbonate transporters are expressed in collecting duct cells, but they do not say which transporters they refer to

Response: Comparison to published literature is now expanded. We discussed in more detail the potential significance of differential RALYL expression, as an example. We now clarify the differential expression of specific bicarbonate transporters in papillary vs. cortico-medullary collecting duct.

It is surprising to see that AQP2 is expressed in intercalated cells. This raises further concerns about cell type annotation (to my knowledge AQP2 is strictly specific to PC).

Response: Thank you for this comment. Knepper and colleagues described a population of “hybrid cells” that express markers of principal and intercalated cells in whole mouse kidneys ². Katalin Suztak’s lab has also showed in mouse kidneys the plasticity of collecting duct cells with a transitional phenotype that express markers from both PC and IC ³. More recently, the Hains-Schwaderer lab showed the existence of these hybrid cells in human kidneys ⁴. Our snRNA seq data show that IC cells in the papilla have more expression of AQP2, and in-situ hybridization

confirms that scattered cells with SLC26A7 expression do have AQP2 RNA (**Supplementary Fig. 1**). These “hybrid” cells are more enriched in the papilla compared to the cortex (**Supplementary Fig.1**). Therefore, our data suggest that the papillary IC population likely comprises a subpopulation with exclusive expression of IC markers, and a larger population with a transitional (“hybrid”) phenotype comprising markers overlapping with PapPC cells.

Data need to be made available to the community beyond this manuscript, e. g. via web interface or public databases.

Response: We agree, and this is our goal. We clarified in the data availability statement and in **Supplementary Table 2** the GEO accessions for the transcriptomics data. The source imaging data will also be available through Zenodo repository: doi: 10.5281/zenodo.7653239

The title of the manuscript ("A spatially anchored transcriptomic atlas of the human kidney papilla identifies significant immune injury and matrix remodeling in patients with stone disease") is somewhat misleading, since no evidence of matrix remodeling is presented (only differential expression of MMPs).

Response: We changed the title to: "A spatially anchored transcriptomic atlas of the human kidney papilla identifies significant immune injury in patients with stone disease".

Minor points:

Abbreviations are used inconsistently. E. g. beta-catenin is called “b-catenin” in Suppl. Table 3, but CTNNB1 in Fig. 1U. Osteopontin is referred to as OPN, SSP1 (Supp. Table 3) and SPP1 (Main text and Fig. 1Q/W). This makes the paper very hard to read.

Response: We apologize for confusion and standardized the abbreviations to be consistent with gene name.

Fig. 1: It should be clearly stated, which of the panels refer to snRNA-seq, ST and CODEX.

Response: We added additional info to the legend to make it clearer, as suggested. We also restructured the results and data presentation so that data from each technology is clearly separated.

Fig. 1W: The authors claim to have identified interstitial cells in the papilla and to have divided them into separate cell classes based on unsupervised analyses. However, no data are shown to support this. Fig. 1W simply shows violin plots of 6 selected markers.-

Response: We extended the data in additional panels now in **Figure 2**, that show the cell clustering and examples of each cell type in the interstitium. We also enlarged the previous panels and added more information to show the cell types identified.

Fig. 2F: For some cell types, no data appear to be available in stone carriers (IC, TAL). Please explain

Response: Because the limited numbers of samples previously, some rarer cell types in the papilla were not fully represented. As we expanded the number of samples, we now see a representation of all papillary cell types.

Fig. 6: It is unclear how Fig. 6 connects with the rest of the paper. The authors show phospho-cJUN-positive cells and claim a connection to ROS. What is the identity of these cells? Why was this done and how does it support the conclusions of this study?

Response: **Figure 6** extends and validates the findings of immune activation and oxidative injury to tissue replicates (at the individual level) without large mineral deposits using another orthogonal technology: large scale 3D imaging and tissue cytometry. c-JUN activation was not restricted to a specific cell type (cells morphologically consistent with collecting ducts, thin limbs and interstitial cells had positive staining), which is consistent with the transcriptomics data. C-JUN activation is known to be induced by oxidative injury (examples ^{5, 6}).

Fig. 7: Please indicate N per group and statistical tests used in figure legend.

Response: these were added as suggested.

Fig. 1Q: *** denotes $p < 0.01$ □ Which statistical test was used?

Response: We added they test in the figure legend –

Reviewer 2:

General response: Thank you for the encouraging comments about the strength of the study. We have now increased the number of samples and added new validation, as outlined above. We made the number samples and the analysis performed transparent in **Supplementary Table 2**. The expanded data solidified our findings.

As a reviewer, I appreciate how hard it is to obtain these rare patient samples. On the other hand, the conclusions being made by the authors based on these small numbers of samples are not really supported by the data. Additional samples need to be used for validation, using appropriate orthologous methods, including in situ hybridization and/or more targeted immunofluorescence studies to draw these conclusions.

Response: Thank you for these suggestions. As discussed above, in addition to increasing the number of samples, we performed in situ hybridization and targeted IF to validate the conclusions.

In addition, much more information needs to be provided about validation and cell type specificity of antibodies used in the CODEX studies, and criteria used based on these markers to define subpopulations of cells (epithelial and macrophages) since it is not apparent in the manuscript.

Response: Thank you for this comment and apologies for the brevity in the previous version. We now added additional details in the methods and provided new **Supplementary Figs. 6, 7 and 8** to show the profile of each cell type identified by this process.

The authors should also provide information about the reason for the nephrectomies, and whether papilla samples were tumor free, and the clinical indication for ureteroscopy (for example whether obstructed)

Response: We now specified the reason for nephrectomies (either tumor nephrectomies with clear margins or transplant donor nephrectomies). The percutaneous nephrolithotomies were elective, without acute obstruction. This data was also added to **Supplementary Table 2**.

Histological analysis needs to clarify how and who identified mineralized regions of the samples

Response: mineralized regions were identified by papilla expert Jim Williams. In fluorescence imaging, we relied on autofluorescence properties of mineral deposits that we described previously ⁷. This has been now added to methods.

Overlay with histology. Are these the same samples or sequential sections? If these are sequential sections, how were the images registered to allow for differences between sections?

Response: Overlay with histology was performed on the same section following the 10x Visium protocol. For CODEX, we also perform H&E staining on the same section after CODEX is performed.

CODEX but no details provided about whether AB localization was compared with other antibodies raised against the same protein, QC validation sections etc., validation after different numbers of CODEX cycles etc.

Response: We added a new section in the methods and data describing the quality control of CODEX, antibody validation and also aligning with histological landmarks, and new supplementary figures for CODEX quality control, as discussed above.

Provide glossary of abbreviations used in figure legends

Response: we now added a new **Supplemental Table 1** with all abbreviations used in figures. The list of antibody targets is shown in **Supplementary Table 3**.

Histology/ST overlays in F/I/L and O are almost impossible to see and need to be accentuated. Better would be to overlay with MxIF/CODEX to properly identify and validate results.

Response: We now added feature plots to better show alignments of cell types with histological structures and expanded the ST histological data with better annotation. In addition, we have added new IF data (**Supplementary Figs. 9 and 12**) that validates and support the inferences from ST /histology overlays.

Much more information needs to be provided to explain how the anticipated cell types were identified by CODEX using 32 markers, many of which are non-specific (e.g.: Krt8, which is touted as CD specific (Figure 1 U) but in fact is pan-epithelial, and while is strongly co-localizes with AQP2, also co-stains with AQP1, and LRP2, LTL, Uromodulin, SLC12A3 and Calbindin).

Response: We added more information in the methods about assignment of cell identities to the various clusters depicted by the analysis and showed specific examples in new **Supplementary Figs. 7-8**. We agree that the CODEX data did not have specific conventional cortical CD markers. However, the combination of the various markers used in the papilla, the staining profile shown in **Figure 2** and **Supplementary Figs. 7-8**, and the mapping to morphologically identifiable CD (validated by orthogonal AQP2 staining in IF (**Supplementary Figs. 9 and 12**)) allowed us to reliably assign an identity for the cell types. We also performed additional testing of the sensitivity and specificity of our methods, as requested by Reviewer 3 (below).

The authors also need to comment on markers used to define macrophage subpopulations, since from list in S Table 3, the numbers of markers are small (CD68, CD206, and perhaps HLA-DR). Moreover, the authors state in S Figure 3 and in the text that CD68 is a marker of

“activated” and “inflammatory” macrophages. This is incorrect; CD68 is pan-monocyte/macrophage marker

Response: Thank you for this comment. We acknowledge that the definition of macrophage is evolving and there is overlap in markers. In addition, the definitions in humans may be different than animal models, and markers for staining macrophages in human tissue are limited. We used here: CD68, CD206, HLA-DR and CD11c, along with pan leukocyte marker CD45, and also the absence of markers for lymphocytes CD3, CD4, CD8, CD20, FOXP3, CD45RO. Regarding CD68, we apologize for the confusion. Although it has been reported to be present in all monocyte/macrophages at various levels, the expression of CD68 is induced by inflammation^{8, 9, 10, 11}, and high expression is typically reported around inflammatory processes in the tissues (tumor or infectious nidus)⁸. Therefore, it is reasonable, and consistent with the literature to associate macrophages with high CD68 expression (based on mean intensity fluorescence and unsupervised analysis) as linked to inflammation. We also find that these cells have a high expression of CD11c, which is also consistent with published data suggesting that inflammatory or M1 macrophages have high expression of CD11c^{12, 13}. This is now further clarified in the text. Many of the macrophages with low CD68 expression that are not associated with inflammation also express CD206. This is consistent with recent data from McEvoy et al showing that CD206+ cells are a predominant population in healthy kidneys¹⁴. We used the various levels of CD68, CD11C and CD206 to classify macrophage subtypes. We now added in the limitation that this approach has its drawbacks, but it is supported by the literature and available tools for spatially defining immune cells in human kidney tissue.

Color coding for a number of the multiplex images is not clear. Also staining with Thy1, which is not included in the list of CODEX Abs in S Table 3. Likewise no color apparent in Fig 1X

Response: We acknowledge the difficulties in displaying 6-7 colors in a figure, which would impact its clarity. We now included examples of staining for each marker in the new **Supplementary Fig. 7**. THY1 was referred to as CD90 in **Supplementary Table 3**, and this has been clarified.

ST validation of MMP7 in CD lacks resolution and ability to differentiate CD from non-CD structures: a) histology provided does not differentiate CD from other epithelium (ideally this should be by IF); b) ST lacks resolution to ID cell types from these overlays anyway. Some form of multiplexed in situ hybridization in other samples, not used for ST analysis, would be appropriate here

Response: As suggested, we now added IF panels to support the inferences from ST /histology overlays (Now **Supplementary Fig. 12**).

Likewise they use ST in mineralized vs. non mineralized and contiguous regions showing increase in certain genes in contiguous regions, focusing on MMP9, CHIT1 and SSP1 by ST. This is appropriate for ST spatial resolution but should also be validated on additional samples using an orthologous method (e.g.: ISH).

Response: We now added codex data (new panels J-O in **Figure 4**) showing the association of SPP1 (uncovered by the ST data) with the mineralized areas.

Reviewer 3:

General response: We thank the reviewer for the overall supportive comments.

I understand one of the limitations of this study was difficulty in obtaining human kidney papilla biopsy specimens. Would the author explain in how many patients and normal posses enough samples to do spatial transcriptomic study and single nuclear RNA sequencing

Response: Thank you for this comment. We added more samples, as discussed above, to solidify the findings that we reported. It is difficult to estimate what is the optimal number of samples needed for the assays. The depth of interrogation, particularly with orthogonal validations using various technologies showing consistent results give high confidence in our conclusions. Of course, single nuclear RNA sequencing with > 75,000 cells from the papilla gives power for our analysis.

would the authors explain the specificity and sensitivity of high resolution large-scale multiplexed 3D and Co-Detection by indexing (CODEX).

Response: Thank you for this question. To assess the efficacy of cell-type detection using 3D tissue cytometry (3DTC) and CO-Detection by InDEXing (CODEX), the specificity and sensitivity were calculated using the manual image-based labeling function of the VTEA software (<https://vtea.wiki>). Forty to fifty cells were randomly selected from each tissue and labeled by a renal expert to generate a ground truth which were compared to the unsupervised classification strategies G-means or Ward hierarchical clustering for 3DTC and CODEX imaging data, respectively. Detection of

MMP7+ collecting duct cells was tested in the 3DTC data, while the detection of all collecting duct cells was tested in the CODEX data. As shown in the presented graph, both imaging based technologies have high sensitivity and specificity (which could be explained by being antibody-based, and by the rigor of the analytical pipeline devised). Key points of this discussion is now added to the methods section.

As authors clearly indicated, the cellular mechanisms described elegantly are global and as they alluded to, such as global event would not provide sufficient knowledge toward limited cell niches associated with mineral deposition in the renal papilla.

Response: Thank you for this comment. We agree that stone disease induces cellular changes across multiple cell types in the papilla. Yet, we show that mineral deposition is associated with an active immune response that spans from inflammation to fibrosis. The transition of epithelial cells, particularly thin limbs, toward an undifferentiated/injured phenotype is in agreement with previous reports by Evan and colleagues that mineral deposition (and possibly injury) starts locally in the thin limbs ¹⁵.

Although the manuscript is very well written, it is extremely heavy in text, specifically in the discussion section. I highly recommended specific attention to be made in summarizing the discussion.

Response: We condensed and reduced the size of the discussion as suggested.

References for the response

1. Lake BB, *et al.* An atlas of healthy and injured cell states and niches in the human kidney. *bioRxiv*, 2021.2007.2028.454201 (2021).
2. Chen L, *et al.* Transcriptomes of major renal collecting duct cell types in mouse identified by single-cell RNA-seq. *Proc Natl Acad Sci U S A* **114**, E9989-E9998 (2017).
3. Park J, *et al.* Single-cell transcriptomics of the mouse kidney reveals potential cellular targets of kidney disease. *Science* **360**, 758-763 (2018).
4. Saxena V, *et al.* Kidney intercalated cells are phagocytic and acidify internalized uropathogenic Escherichia coli. *Nat Commun* **12**, 2405 (2021).
5. Kaneto H, Xu G, Fujii N, Kim S, Bonner-Weir S, Weir GC. Involvement of c-Jun N-terminal kinase in oxidative stress-mediated suppression of insulin gene expression. *J Biol Chem* **277**, 30010-30018 (2002).
6. Osto E, *et al.* c-Jun N-terminal kinase 2 deficiency protects against hypercholesterolemia-induced endothelial dysfunction and oxidative stress. *Circulation* **118**, 2073-2080 (2008).
7. Winfree S, *et al.* Multimodal imaging reveals a unique autofluorescence signature of Randall's plaque. *Urolithiasis* **49**, 123-135 (2021).
8. Chistiakov DA, Killingsworth MC, Myasoedova VA, Orekhov AN, Bobryshev YV. CD68/macrosialin: not just a histochemical marker. *Lab Invest* **97**, 4-13 (2017).
9. Rabinowitz SS, Gordon S. Macrosialin, a macrophage-restricted membrane sialoprotein differentially glycosylated in response to inflammatory stimuli. *J Exp Med* **174**, 827-836 (1991).
10. Ramprasad MP, Terpstra V, Kondratenko N, Quehenberger O, Steinberg D. Cell surface expression of mouse macrosialin and human CD68 and their role as macrophage receptors for oxidized low density lipoprotein. *Proc Natl Acad Sci U S A* **93**, 14833-14838 (1996).
11. Yoshida H, Quehenberger O, Kondratenko N, Green S, Steinberg D. Minimally oxidized low-density lipoprotein increases expression of scavenger receptor A, CD36, and macrosialin in resident mouse peritoneal macrophages. *Arterioscler Thromb Vasc Biol* **18**, 794-802 (1998).

12. Lumeng CN, DelProposto JB, Westcott DJ, Saltiel AR. Phenotypic switching of adipose tissue macrophages with obesity is generated by spatiotemporal differences in macrophage subtypes. *Diabetes* **57**, 3239-3246 (2008).
13. Zhu Y, *et al.* Identification of different macrophage subpopulations with distinct activities in a mouse model of oxygen-induced retinopathy. *Int J Mol Med* **40**, 281-292 (2017).
14. McEvoy CM, *et al.* Single-cell profiling of healthy human kidney reveals features of sex-based transcriptional programs and tissue-specific immunity. *Nat Commun* **13**, 7634 (2022).
15. Evan AP, *et al.* Randall's plaque of patients with nephrolithiasis begins in basement membranes of thin loops of Henle. *J Clin Invest* **111**, 607-616 (2003).

REVIEWERS' COMMENTS

Reviewer #1 (Remarks to the Author):

The authors have adequately responded to most of my comments. Additional samples were added. The amount of presented data is impressive. The main strength of this manuscript is the fact that it provides a data resource associated with kidney stone disease. It is limited by the fact that the findings and conclusions are descriptive.

Reviewer #2 (Remarks to the Author):

I am satisfied that this revision has addressed my most significant concerns, in particular by increasing sample numbers (I am sure, quite an effort!) and clarifying the source of tissues and assays performed on each in a revised and expanded S. Table 2.

We thank the reviewers for their input. Below is a response to the comments.

Reviewer #1 (Remarks to the Author):

The authors have adequately responded to most of my comments. Additional samples were added. The amount of presented data is impressive. The main strength of this manuscript is the fact that it provides a data resource associated with kidney stone disease. It is limited by the fact that the findings and conclusions are descriptive.

We thank the reviewer for these comments and the enthusiasm for the added samples and revisions. We agree that this work does not have an experimental intervention (consistent with what is expected from working with patient samples). The reviewer pointed out a major strength of this work as a resource for the community. In addition, understanding the cellular and molecular events that occur in human stone disease and uncovering the landscape of the human papilla at high resolution (as was done in this work), are necessary to understand the pathogenesis of nephrolithiasis and devise experimental strategies and interventions that have translational implications on patients. Our work also delivers unique opportunities to discover biomarkers (such as MMP7 and MMP9) that could impact diagnosis and management of stone disease.

Reviewer #2 (Remarks to the Author):

I am satisfied that this revision has addressed my most significant concerns, in particular by increasing sample numbers (I am sure, quite an effort!) and clarifying the source of tissues and assays performed on each in a revised and expanded S. Table 2

We thank the reviewer for these comments, and we appreciate this feedback!